# Structural mechanism of ligand activation in human calcium-sensing receptor

Yong Geng[1,2], Lidia Mosyak[1], Igor Kurinov[3], Hao Zuo[1], Emmanuel Sturchler[4], Tat Cheung Cheng[1], Prakash Subramanyam[5], Alice P Brown[6], Sarah C Brennan[6], Hee-chang Mun[6], Martin Bush[1], Yan Chen[1], Trang X Nguyen[7], Baohua Cao[1], Donald D Chang[5], Matthias Quick[7], Arthur D Conigrave[6], Henry M Colecraft[5], Patricia McDonald[4], Qing R Fan[1,8]*

[1]Department of Pharmacology, Columbia University, New York, United States; [2]Shanghai Institute of Materia Medica, Chinese Academy of Sciences, Shanghai, China; [3]Department of Chemistry and Chemical Biology, Cornell University, Ithaca, United States; [4]Department of Molecular Therapeutics, The Scripps Translational Science Institute, Jupiter, United States; [5]Department of Physiology and Cellular Biophysics, Columbia University, New York, United States; [6]School of Life and Environmental Sciences, University of Sydney, New South Wales, Australia; [7]Department of Psychiatry, Columbia University, New York, United States; [8]Department of Pathology and Cell Biology, Columbia University, New York, United States

*For correspondence: qf13@
cumc.columbia.edu

**Competing interests:** The authors declare that no competing interests exist.

**Abstract** Human calcium-sensing receptor (CaSR) is a G-protein-coupled receptor (GPCR) that maintains extracellular $Ca^{2+}$ homeostasis through the regulation of parathyroid hormone secretion. It functions as a disulfide-tethered homodimer composed of three main domains, the Venus Flytrap module, cysteine-rich domain, and seven-helix transmembrane region. Here, we present the crystal structures of the entire extracellular domain of CaSR in the resting and active conformations. We provide direct evidence that L-amino acids are agonists of the receptor. In the active structure, L-Trp occupies the orthosteric agonist-binding site at the interdomain cleft and is primarily responsible for inducing extracellular domain closure to initiate receptor activation. Our structures reveal multiple binding sites for $Ca^{2+}$ and $PO_4^{3-}$ ions. Both ions are crucial for structural integrity of the receptor. While $Ca^{2+}$ ions stabilize the active state, $PO_4^{3-}$ ions reinforce the inactive conformation. The activation mechanism of CaSR involves the formation of a novel dimer interface between subunits.

## Introduction

The extracellular calcium-sensing receptor (CaSR) is responsible for the maintenance of a stable blood $Ca^{2+}$ level (*Brown et al., 1993*; *Hofer and Brown, 2003*). It senses fluctuations in the circulating $Ca^{2+}$ concentration and controls $Ca^{2+}$ homeostasis by (1) modulating the production of parathyroid hormone in parathyroid glands, and (2) regulating the reabsorption of $Ca^{2+}$ in kidney and bone (*Brown, 2013*). Abnormal function of CaSR is associated with $Ca^{2+}$ homeostatic disorders (*Brown, 2007*; *Hendy et al., 2009*; *Ward et al., 2012*). Loss-of-function mutations in CaSR lead to potentially fatal neonatal severe primary hyperparathyroidism, while gain-of-function mutations cause autosomal dominant hypocalcemia (*Hendy et al., 2009*; *Ward et al., 2012*). CaSR also plays important roles in biological processes unrelated to $Ca^{2+}$ balance, including fetal development

**eLife digest** Calcium ions regulate many processes in the human body. The calcium-sensing receptor, called CaSR, is responsible for maintaining a stable level of calcium ions in the blood. This receptor can detect small changes in the concentration of calcium ions, and activates signalling events within the cell to restore the level of calcium ions back to normal. Abnormal activity of this receptor is associated with severe diseases in humans

CaSR is found in the surface membrane of cells and belongs to a family of proteins called G-protein coupled receptors. Much of the protein extends out of the cell and interacts with calcium ions, phosphate ions and certain other molecules such as amino acids. However, it was not well understood how these small molecules bind to CaSR and how this activates the receptor.

Geng et al. have now used a technique called X-ray crystallography to view the three-dimensional structure of the exterior domain of CaSR in its resting state and active state. These structures revealed that, contrary to expectations, calcium ions are not the main activator of the receptor. Instead, Geng et al. found that CaSR adopts an inactive state in the absence or presence of calcium ions, while the active state only forms when an amino acid is bound.

Furthermore investigation showed that calcium ions are needed to stabilise the active form, while phosphate ions keep the inactive form stable. Geng et al. also identified the shape changes that must occur as CaSR transitions from its inactive to its active state. In particular, an amino acid binding to the exterior domain causes it to close like a venus flytrap, which is a crucial step in activating the receptor.

Taken together, the findings show that the amino acids and calcium ions act jointly to fully activate CaSR. The next steps are to determine the structure of the entire receptor with and without its small molecule partners and to use these structures to design drugs that can alter CaSR's activity in order to treat human diseases.

---

(*Riccardi et al., 2013*), nutrient sensing (*Conigrave and Hampson, 2006*), and regulation of neuronal excitability (*Ruat and Traiffort, 2013*).

The functional diversity of CaSR results from its ability to activate multiple signaling pathways through $G_{q/11}$, $G_{i/o}$, $G_{12/13}$ and $G_s$ proteins (*Conigrave and Ward, 2013*; *Hofer and Brown, 2003*; *Magno et al., 2011*), and to respond to a variety of ligands (*Magno et al., 2011*; *Saidak et al., 2009*). The general consensus is that the principal agonist of CaSR is extracellular $Ca^{2+}$ (*Hofer and Brown, 2003*). Other orthosteric agonists include various divalent and trivalent cations, polyamines and cationic polypeptides (*Magno et al., 2011*; *Saidak et al., 2009*).

CaSR function can be regulated by endogenous and synthetic allosteric modulators, extracellular pH and ionic strength (*Jensen and Brauner-Osborne, 2007*; *Quinn et al., 2004*, *1998*; *Saidak et al., 2009*). Aromatic and aliphatic L-amino acids such as L-Phe and L-Trp increase the sensitivity of CaSR toward $Ca^{2+}$ (*Conigrave et al., 2000*) and are considered as positive allosteric modulators of the receptor (*Saidak et al., 2009*). Previous studies have also demonstrated that L-amino acids can activate the receptor provided that $Ca^{2+}$ concentration is above a threshold (*Conigrave et al., 2004*; *Conigrave et al., 2000*; *Rey et al., 2005*; *Young and Rozengurt, 2002*). For this reason, L-amino acids have been called allosteric activators (*Conigrave et al., 2004*). Finally, it has been proposed that $Ca^{2+}$ ions and amino acids may act as co-agonists of the receptor (*Conigrave et al., 2000*; *Young and Rozengurt, 2002*). However, this view has not been widely accepted.

CaSR is a class C G-protein-coupled receptor (GPCR). Within this family, CaSR and metabotropic glutamate receptors (mGluRs) function as disulfide-linked homodimers (*Bai et al., 1998*; *Pidasheva et al., 2006*; *Ray et al., 1999*; *Romano et al., 1996*; *Ward et al., 1998*; *Zhang et al., 2001*), while $GABA_B$ and taste receptors are obligatory heterodimers (*Jones et al., 1998*; *Kaupmann et al., 1998*; *Kuner et al., 1999*; *Nelson et al., 2002, 2001*; *Ng et al., 1999*; *White et al., 1998*). Ligand binding to CaSR takes place within a large extracellular Venus Flytrap (VFT) module that consists of two domains (LB1 and LB2) (*Brauner-Osborne et al., 1999*; *Hendy et al., 2013*; *Mun et al., 2004*). In addition, CaSR contains a cysteine-rich (CR) domain that

connects the VFT module to the transmembrane region (*Hendy et al., 2013*). This CR region is present in all class C GPCRs except GABA$_B$ receptor and is required for receptor activation (*Hauache et al., 2000*; *Hu et al., 2000*; *Huang et al., 2011*). However, the activation mechanism of CaSR remains unknown.

Structural information for class C GPCRs is available for the extracellular domains (ECD) of mGluRs (*Kunishima et al., 2000*; *Muto et al., 2007*; *Tsuchiya et al., 2002*) and GABA$_B$ receptor (*Geng et al., 2013*, *2012*), as well as the transmembrane domains of mGluRs (*Dore et al., 2014*; *Wu et al., 2014*). Here, we present the first crystal structures of the entire extracellular domain of human CaSR in two different functional states. These structures reveal novel binding sites for Ca$^{2+}$, PO$_4^{3-}$ and L-Trp, identify L-Trp as an agonist of the receptor, and demonstrate that these ions and amino acids collectively control the function of CaSR.

## Results

### Structure of CaSR ECD homodimer

The ECD of human CaSR was secreted from baculovirus-infected insect cells as a disulfide-tethered homodimer (*Figure 1—figure supplement 1*). It contains 11 potential N-linked glycosylation sites. Disruption of three of the glycosylation sites did not alter CaSR signaling (*Figure 1—figure supplement 1*). Formation of well-diffracting crystals required partial deglycosylation of the receptor through mutation and enzymatic digestion. We obtained two different forms of CaSR ECD crystals. Form I was crystallized in the absence and presence of 2 mM Ca$^{2+}$, and form II in the presence of 10 mM L-Trp and 10 mM Ca$^{2+}$ (*Table 1*).

In both CaSR ECD structures, the two protomers interact in a side-by-side fashion while facing opposite directions (*Figure 1*; *Figure 1—figure supplement 2*). Each CaSR ECD protomer consists of three domains, LB1, LB2 and CR. The two lobe-shaped domains LB1 and LB2 form a VFT module similar to that of mGluRs (*Kunishima et al., 2000*; *Muto et al., 2007*; *Tsuchiya et al., 2002*) and GABA$_B$ receptor (*Geng et al., 2013*, *2012*). The relative orientation between the LB2 and CR domains is fixed through an interdomain disulfide linkage (C236-C561), and the CR domain is positioned to amplify and transmit the conformational variations within the VFT module.

Form I crystal structure of CaSR ECD represents the inactive configuration since the VFT modules of both protomers are in the open conformation associated with the resting state (open-open), and the interdomain cleft is empty. In addition, each protomer structure contains one Ca$^{2+}$ ion and three SO$_4^{2-}$ ions (*Figure 1A*).

In the form II crystal structure, both protomers of CaSR ECD have the closed conformation associated with agonist binding (closed-closed). Surprisingly, the ligand-binding cleft of each protomer is solely occupied by an L-Trp molecule. Ca$^{2+}$ is bound at four novel sites in the CaSR ECD structure, including one at the homodimer interface. Each CaSR ECD molecule also contains two PO$_4^{3-}$ ions (*Figure 1B*).

Agonist binding induces large conformational changes within the CaSR ECD homodimer. First, the VFT module of each protomer undergoes domain closure. Alignment based on the LB1 domains showed that the LB2 domains of inactive and active structures are related by a 29° rotation (*Figure 2A*). Second, the LB2 domains of the two protomers approach each other, resulting in an expansion of the homodimer interactions involving LB2 domains. Third, the CR domains of the two subunits interact to form a large homodimer interface that is unique to the active state. The CR domains are brought into close contact by the motion involving LB2 since the two domains are rigidly associated within each subunit. Finally, the structural reorganization of CR domains reduces the distance between the C-termini of the two subunits from 83 Å to 23 Å (*Figure 2B,C*). This CR domain movement may cause reorientation of the transmembrane domains during receptor activation.

### Common protomer-protomer interactions

The inactive structure of CaSR ECD shows that subunit association in the resting state is primarily mediated by the LB1 domains (*Figure 3A*; *Figure 3—figure supplement 1*). This dimer interface is largely conserved in the active structure, indicating that the LB1-LB1 interaction mostly serves to faciliate dimerization between receptor subunits (*Figure 3B*; *Figure 3—figure supplement 1*).

**Table 1.** Data collection and refinement statistics.

| Functional state | Inactive (2 mM Ca$^{2+}$) | Active (10 mM Ca$^{2+}$, 10 mM L-Trp) |
|---|---|---|
| **Crystal** | Form I | Form II |
| **Data collection** | | |
| Space group | F222 | C2 |
| Wavelength (Å) | 0.9792 | 1.7712 |
| Cell dimensions | | |
| *a, b, c* (Å) | 126.3, 150.2, 214.6 | 107.7, 127.5, 146.8 |
| α, β, γ (°) | 90.0, 90.0, 90.0 | 90.0, 108.7, 90.0 |
| Resolution (Å) | 88.1 - 3.1 (3.6 - 3.1) | 139.0 - 2.6 (2.9 - 2.6) |
| $R_{sym}$ or $R_{merge}$ | 0.051 (0.702) | 0.043 (0.575) |
| $I / \sigma I$ | 21.3 (2.1) | 22.1 (2.3) |
| Completeness (%) | 99.9 (100.0) | 98.0 (97.1) |
| Redundancy | 6.6 (6.8) | 6.9 (6.8) |
| $CC_{1/2}$ (%) | 100.0 (93.1) | 99.9 (96.9) |
| **Refinement** | | |
| Resolution (Å) | 107.2 - 3.1 | 37.5 - 2.6 |
| No. of reflections | 16,747 | 48,839 |
| $R_{work}$ / $R_{free}$ (%) | 22.2 / 23.9 | 21.1 / 22.2 |
| No. of atoms | | |
| Protein | 4564 | 8454 |
| Ligand (Trp) | - | 30 |
| Cation (Ca$^{2+}$) | 1 | 8 |
| Anion | 15 ($SO_4^{2-}$) | 20 ($PO_4^{3-}$) |
| Sugar | 98 | 70 |
| Water | 43 | 331 |
| *B*-factors (Å$^2$) | | |
| Protein | 110.4 | 68.3 |
| Ligand | - | 39.9 |
| Cation (Ca$^{2+}$) | 105.4 | 96.3 |
| Anion | 102.1 ($SO_4^{2-}$) | 61.0 ($PO_4^{3-}$) |
| Sugar | 152.9 | 80.4 |
| Water | 76.9 | 53.8 |
| R.m.s. deviations | | |
| Bond lengths (Å) | 0.008 | 0.009 |
| Bond angles (°) | 1.15 | 1.14 |

Values in parentheses are for highest-resolution shell.

$CC_{1/2}$ is defined in reference (**Karplus and Diederichs, 2012**).

**Source data 1.** Statistics for anomalous data collection.

**Source data 2.** Data collection and refinement statistics for endogenous ligand-bound CaSR ECD.

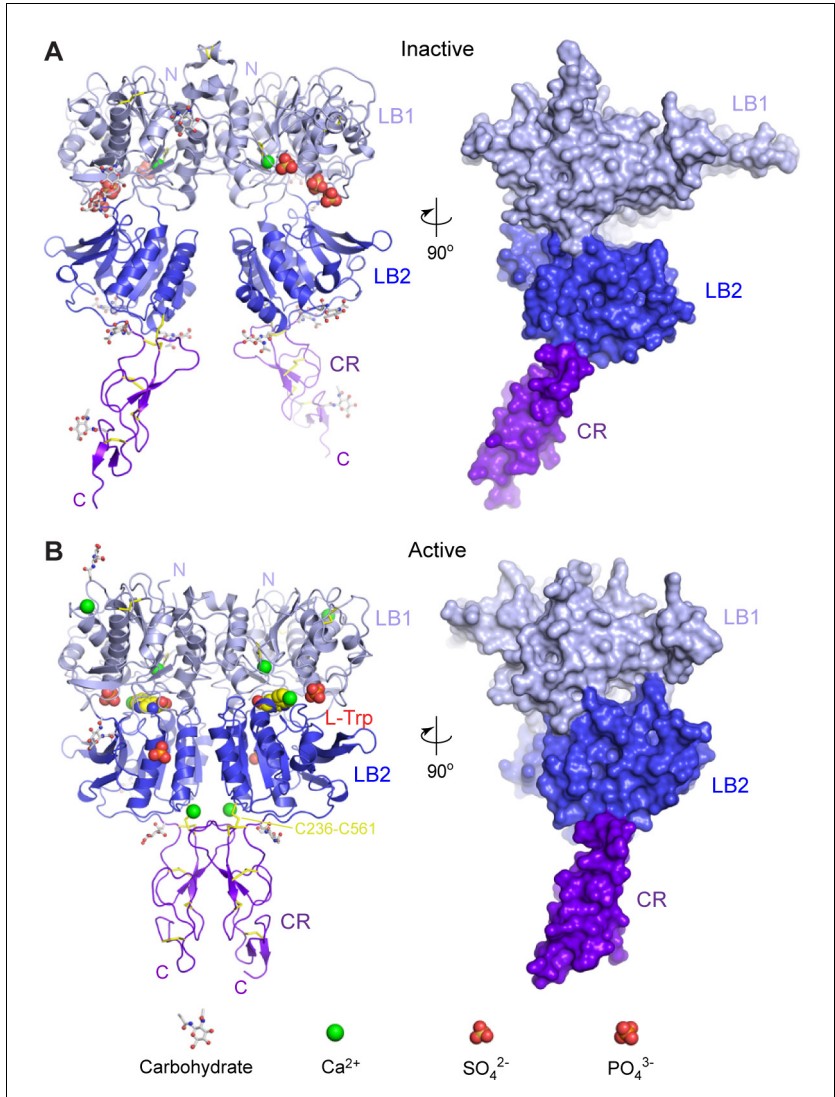

**Figure 1.** Crystal structures of human CaSR ECD. (A) Inactive-state structure of CaSR ECD homodimer in the presence of 2 mM $Ca^{2+}$. (B) Active-state structure of CaSR ECD homodimer in the presence of 10 mM $Ca^{2+}$ and 10 mM L-Trp. Each structure is shown in cartoon (front view) or surface (side view) representations that are related by a 90°-rotation about the vertical axis. Each protomer is colored according to its individual domains (LB1, light blue; LB2, blue; CR, purple). The various ligands (L-Trp, $Ca^{2+}$, $PO_4^{3-}$, $SO_4^{2-}$) are displayed as space-filling models. Observed carbohydrates are shown as ball-and-stick models in gray. Disulfide bridges are in yellow.

The following figure supplements are available for figure 1:

**Figure supplement 1.** Purification of the CaSR ECD homodimer.

**Figure supplement 2.** Different conformational states of CaSR ECD.

The LB1-LB1 dimer interface buries over 3800 Å$^2$ of solvent-accessible surface area, and can be divided into two regions. Site I is located at the center of LB1 domain and is flanked on either side by the two symmetric parts of site II (*Figure 3A,B*).

Site I is formed by two central helices (B, D) of each protomer (*Figure 3A,B*; *Figure 3—figure supplement 1*). A dimer interface formed by the same structural elements is also observed in mGluR and GABA$_B$ structures (*Geng et al., 2013*; *Kunishima et al., 2000*; *Muto et al., 2007*; *Tsuchiya et al., 2002*). For mGluRs, transition between the resting and active configurations causes

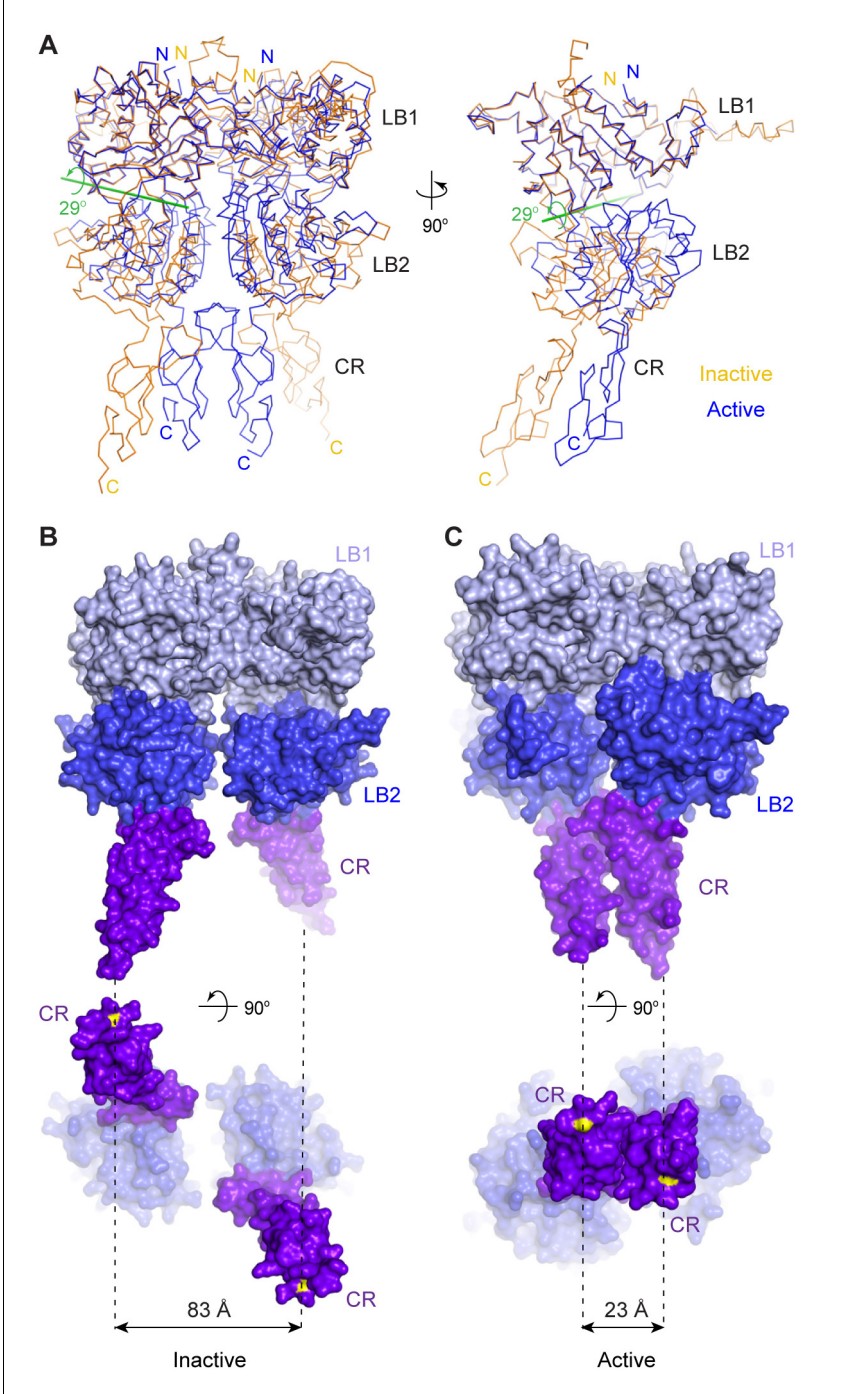

**Figure 2.** Agonist-induced conformational changes. (**A**) Superposition of the inactive (orange) and active (blue) CaSR ECD structures based on the LB1 domain of one protomer (front view, left; side view, right). Green line is the axis of rotation that relates the LB2 domains of the superimposed protomers (rotation $\chi = 29°$, screw translation $\tau_\chi = -2.2$ Å). (**B**, **C**) Surface representation of inactive (**B**) and active (**C**) structures in front (top) and bottom (bottom) views. Distance between C-termini of the two subunits (yellow) is marked by dashed line for each homodimer.

a 70°-rotation in the LB1-LB1 homodimer interface (***Kunishima et al., 2000***) (***Figure 3—figure supplement 1***). For both CaSR and GABA$_B$ receptors (***Geng et al., 2013***), however, agonist binding

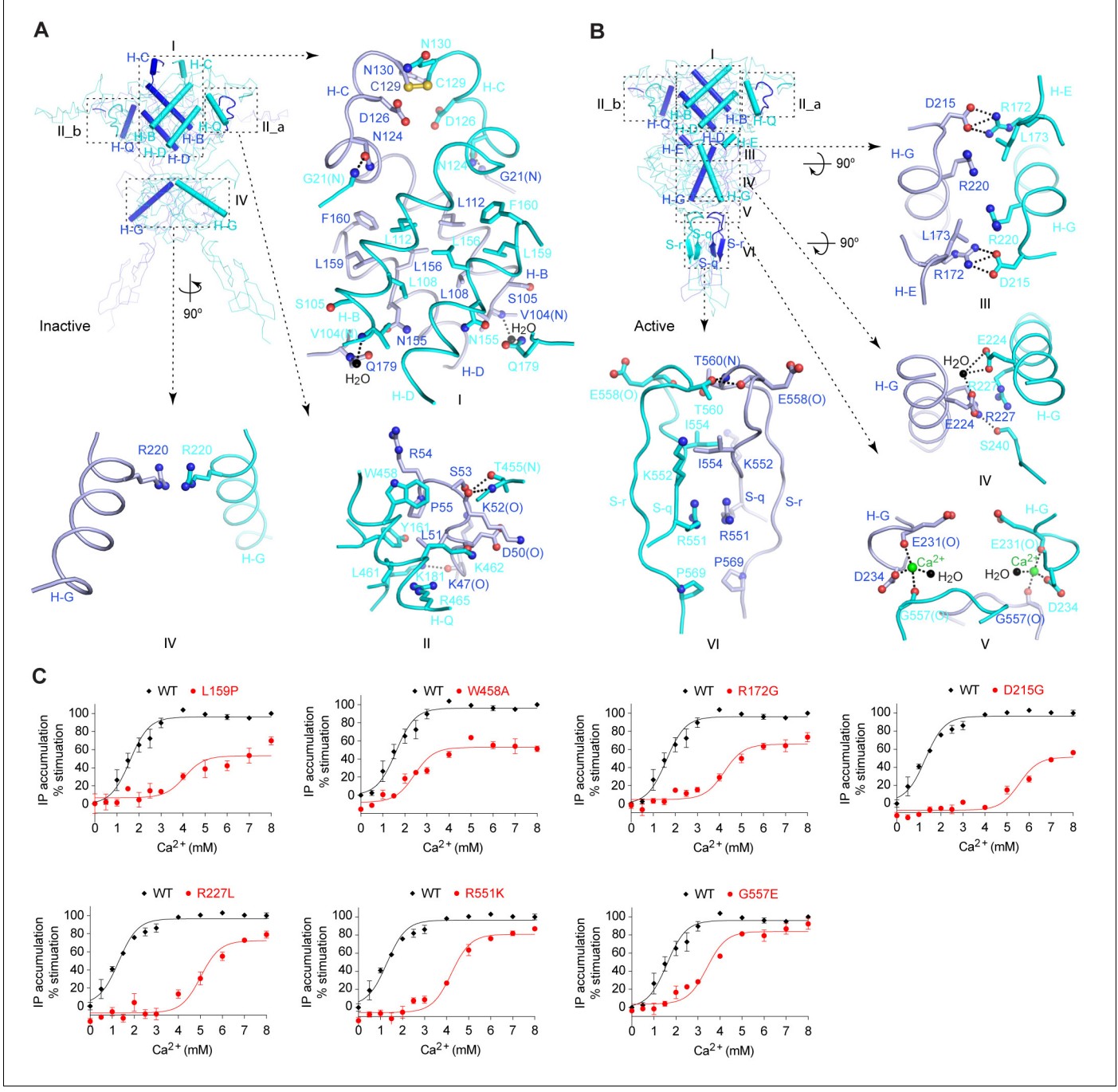

**Figure 3.** Homodimer interface. (**A**) Cα trace of inactive structure with elements involved in homodimer formation highlighted by cartoons. The interface is divided into three regions I, II, and IV. Site II is further separated into two symmetrical parts II_a and II_b. Specific contacts at each interface region are shown in surrounding panels. Dashed lines indicate hydrogen bonds. (**B**) Cα trace of active structure showing elements involved in homodimer formation. The interface is divided into six regions, I, II, III, IV, V, and VI. Specific contacts at the interface areas III, IV, V, and VI are shown in surrounding panels. For both structures, the domains involved in dimerization at each interface region are: I: LB1-LB1; II: LB1-LB1; III: LB2-LB1; IV: LB2-LB2; V: LB2-CR; VI: CR-CR. (**C**) Dose-dependent $Ca^{2+}$-stimulated IP accumulation in cells transiently expressing wild-type (wt) or mutant CaSR. Naturally occurring inactivating mutations L159P, R172G, D215G, R227L, R551K, and G557E are located at the homodimer interface. The single mutation W458A was designed based on structure to affect receptor homodimerization.

The following figure supplements are available for figure 3:

**Figure supplement 1.** Homodimer interface.

*Figure 3 continued*

**Figure supplement 2.** Homodimer interface.

only induces a small 5°-rotation in the orientation of this interface, and the LB1-LB1 dimer interface is largely conserved throughout the activation process (*Figure 3—figure supplement 1*).

The dimer interactions at site I of CaSR are predominantly hydrophobic and involve tightly packed leucine and phenylalanine residues (L112, L156, L159, and F160). The disease-causing mutation L159P renders the receptor less sensitive to Ca$^{2+}$ (*Grant et al., 2011*; *Hendy et al., 2009*) (*Figure 3C*). In addition, site I features an inter-subunit disulfide bridge located at the tip of helix C.

Site II is unique to the CaSR ECD structures. It involves an arm-like long loop stretched out from one subunit to reach its binding partner (*Figure 3A,B*; *Figure 3—figure supplement 1*). The dimer interactions at site II include hydrogen bonds and hydrophobic contacts. Several disease-causing mutations are located at this interface (S53P, P55L, and Y161C) (*Hendy et al., 2009*). Substitution of a deeply buried interfacial residue W458 with alanine also decreased the potency of Ca$^{2+}$ (*Figure 3C*). These observations indicate that formation of a stable homodimer is important for CaSR function.

## Agonist-induced homodimer interface

Agonist binding causes an expansion of the dimer interactions involving LB2 domain. In the inactive homodimer, only minimal contacts occur between the LB2 domains (*Figure 3A*). In the active state, LB2 of one protomer interacts with all three domains of a second protomer (*Figure 3B*). These contacts are predominantly hydrophilic, and bury 1000 Å$^2$ of solvent accessible surface area.

LB2 mediates dimer interactions primarily through a central helix (G) that transverses the domain (*Figure 3B*). First, the top of helix G contacts LB1 domain of the opposing subunit through two symmetric salt bridges between D215 and R172 (site III). Second, the LB2-LB2 contacts involve the midsection of helix G, and feature a hydrogen bond between R227 and S240, and water-mediated contacts by E224 (site IV). Finally, LB2 interacts with the CR domain of a second subunit through a Ca$^{2+}$ ion (site V). This Ca$^{2+}$ ion bridges the end of helix G in LB2 with a loop in CR domain. An abundance of disease-related mutations are found at these dimer interaction sites, including (1) R172G and D215G at site III, (2) R227L and R227Q at site IV, and (3) G557E at site V (*Hendy et al., 2009*). All these mutations disrupt agonist-dependent dimer contacts, and decreased agonist efficacy (*Heath et al., 1996*; *Hendy et al., 2009*; *Wystrychowski et al., 2005*) (*Figure 3C*).

Agonist binding also induces the formation of a novel homodimer interface between the CR domains that covers approximately 1200 Å$^2$ of solvent accessible surface area (site VI) (*Figure 3B*). The CR-CR interactions are mediated by two β-strands and their connecting loop from each subunit. Key contacts include two cross-subunit hydrogen bonds (T560, E558), hydrophobic contacts (I554, P569), and electrostatic interactions (R551). Among these, R551K is a known disease-causing mutation that reduced the receptor response (*Hendy et al., 2009*; *Toke et al., 2007*) (*Figure 3C*).

## Amino acid recognition

In the active structure, the amino acid L-Trp is bound at the interdomain cleft of the VFT module, in agreement with previous mutational data (*Mun et al., 2005*; *Mun et al., 2004*; *Zhang et al., 2014*, *2002*) (*Figure 4A*; *Figure 4—figure supplement 1*). L-Trp facilitates extracellular domain closure of CaSR by contacting both LB1 and LB2 domains of the VFT module (*Figure 4B,C*; *Figure 4—figure supplement 1*). The interactions between CaSR ECD and L-Trp are primarily mediated by hydrogen bonds. (1) The carboxylic acid group of L-Trp forms hydrogen bonds through both oxygen atoms with LB1 residues S147, A168, and S170. (2) The backbone nitrogen of L-Trp is hydrogen-bonded to A168 and S170 of LB1 domain. (3) The indole nitrogen of L-Trp forms two hydrogen bonds with E297 of LB2 domain. (4) L-Trp is engaged in hydrophobic contacts with both LB1 and LB2 residues including W70, T145, Y218, and A298. The extensive contacts between backbone atoms of L-Trp and the receptor suggest that other amino acids may bind to CaSR in a similar fashion to induce domain closure.

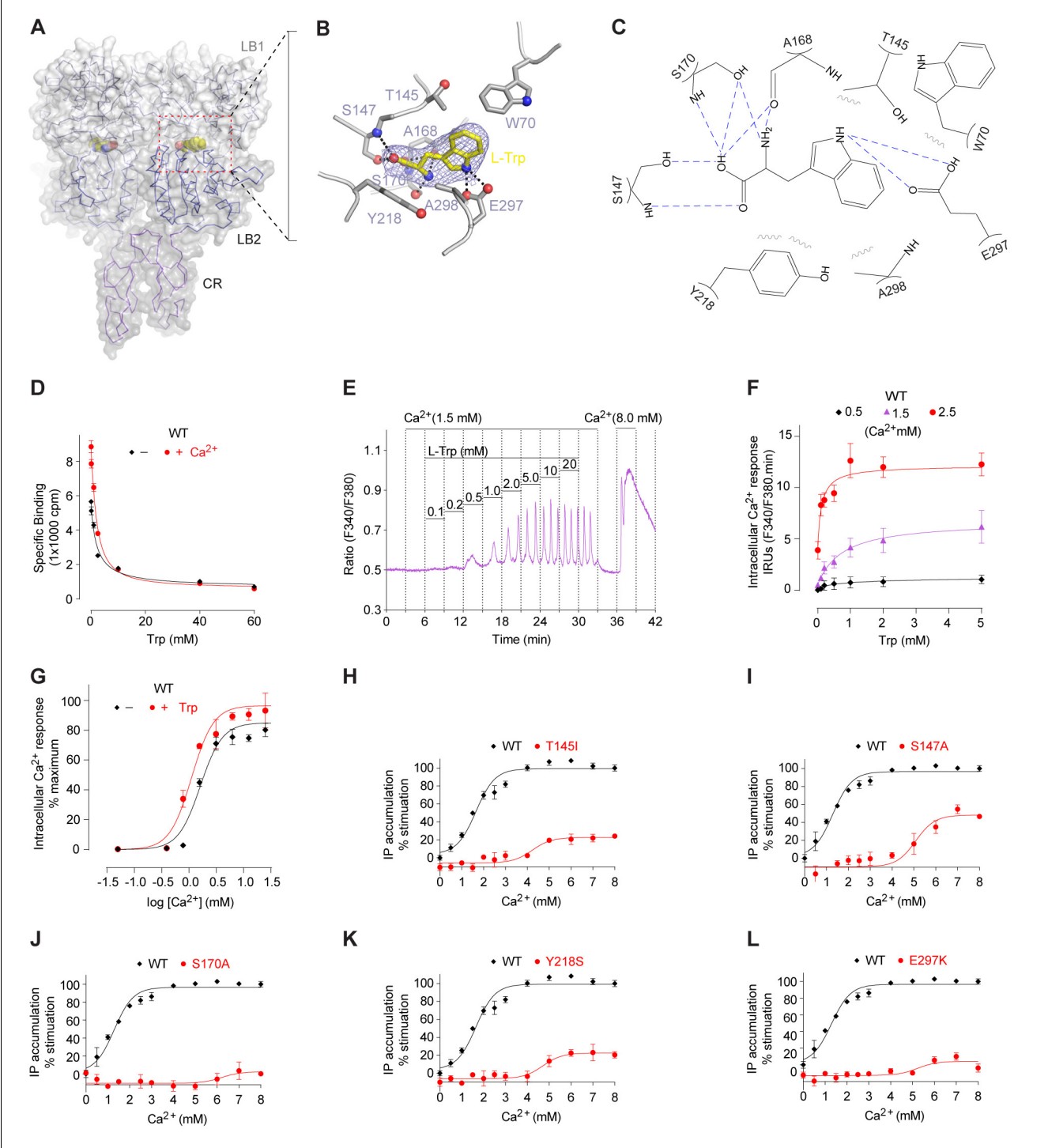

**Figure 4.** L-Trp recognition by CaSR ECD. (**A**) Molecular surface of a L-Trp-bound CaSR ECD protomer in the active structure. L-Trp is displayed as a space-filling model. (**B**) Specific contacts between CaSR ECD (gray) and L-Trp (yellow). Mesh represents the final 2Fo-Fc electron density map contoured at 1σ. Hydrogen bonds are represented by dashed lines. (**C**) Schematic diagram of the interactions between CaSR ECD and bound L-Trp. Selected contacts are highlighted; hydrogen bonds, dashed lines; hydrophobic contacts, wiggled lines. (**D**) Dose-response curves of nonradioactive L-Trp inhibiting [$^3$H]-L-Trp binding to CaSR ECD in the presence of 0 mM or 2 mM Ca$^{2+}$. (**E–F**) Dose-dependent L-Trp-induced intracellular Ca$^{2+}$ mobilization at various extracellular Ca$^{2+}$ concentrations in cells stably expressing CaSR. (**E**) Effect of increments of L-Trp (final concentrations: 0.1, 0.2, 0.5, 1.0, 2.0, 5.0, 10, 20 mM) in the presence of 1.5 mM extracellular Ca$^{2+}$. The experiments were followed by an exposure to 0.5 mM extracellular Ca$^{2+}$ to demonstrate reversibility of the L-Trp-induced intracellular Ca$^{2+}$ oscillation, and a later exposure to 8 mM extracellular Ca$^{2+}$ to demonstrate the maximal response. (**F**) Integrated response curves of L-Trp at 0.5, 1.5, and 2.5 mM Ca$^{2+}$. Intracellular Ca$^{2+}$ responses are presented in integrated

*Figure 4 continued on next page*

*Figure 4 continued*

response units (IRUs, F340/F380.min). (**G**) Effect of L-Trp on $Ca^{2+}$-stimulated intracellular $Ca^{2+}$ mobilization in cells transiently expressing wt CaSR. (**H–L**) Dose-dependent $Ca^{2+}$-stimulated IP accumulation in cells transiently expressing wild-type or mutant CaSR. Naturally-occurring inactivating mutations Y218S and E297K are located at the L-Trp binding site. The single mutations T145I, S147A, and S170A were designed based on structure to disrupt L-Trp recognition.

The following figure supplements are available for figure 4:

**Figure supplement 1.** L-Trp recognition by CaSR ECD.

**Figure supplement 2.** Endogenous ligand of CaSR.

We measured the direct interaction between L-Trp and CaSR ECD by scintillation proximity assay (SPA) (*Quick and Javitch, 2007*) (*Figure 4D*). CaSR ECD exhibited binding of [³H]-L-Trp in the absence and presence of $Ca^{2+}$. Isotope dilution of [³H]-L-Trp with nonradioactive L-Trp led to a displacement of [³H]-L-Trp in a concentration-dependent manner. The addition of 2 mM $Ca^{2+}$ increased the amount of L-Trp bound to CaSR ECD at any given concentration, suggesting that extracellular $Ca^{2+}$ enhances L-Trp binding, possibly by affecting the L-Trp-binding affinity and kinetics of CaSR. The half-maximal inhibitory concentration of L-Trp ($IC_{50}$, concentration at which 50% displacement of [³H]-L-Trp was observed) was approximately 2 mM regardless of whether the experiment was performed in the absence (2.11 ± 0.72 mM) or presence (2.04 ± 0.10 mM) of 2 mM $Ca^{2+}$. Nevertheless, the binding affinity of L-Trp to CaSR ECD with and without $Ca^{2+}$ may still differ as it depends on the concentration and binding affinity of the radiolabeled ligand. Further studies are needed to characterize the effect of $Ca^{2+}$ on L-amino acid binding at the orthosteric agonist site of CaSR.

We found that L-Trp directly stimulated intracellular $Ca^{2+}$ mobilization through CaSR (*Figure 4E, F*), in agreement with previous findings (*Conigrave et al., 2004*, *2000*; *Rey et al., 2005*; *Young and Rozengurt, 2002*). L-Trp-induced CaSR activation required the presence of extracellular $Ca^{2+}$ above a threshold level of 0.5 mM. The effect of L-Trp on CaSR was concentration-dependent, with an apparent half-maximal effective concentration ($EC_{50}$) of 0.12 ± 0.06 mM when extracellular $Ca^{2+}$ was present at 2.5 mM. The efficacy and potency of L-Trp decreased at lower concentrations of $Ca^{2+}$ that are within the physiological range (L-Trp $EC_{50}$ = 0.75 ± 0.51 mM at 1.5 mM $Ca^{2+}$), consistent with previous proposal that multiple amino acids need to act in concert to control the function of CaSR (*Conigrave et al., 2000*, *2004*).

L-Trp is also important for $Ca^{2+}$-stimulated CaSR response. First, L-Trp elevated the sensitivity of CaSR toward extracellular $Ca^{2+}$ (*Conigrave et al., 2000*) (*Figure 4G*). The presence of 10 mM L-Trp lowered the $EC_{50}$ of extracellular $Ca^{2+}$ by about 30%. Second, the residues involved in L-Trp binding are crucial for $Ca^{2+}$-dependent receptor activation. Previous and current studies demonstrate that each of the individual mutations S147A, S170A, Y218A, and E297K abolishes $Ca^{2+}$-induced receptor response (*Silve et al., 2005*; *Zhang et al., 2002*) (*Figure 4H–L*). Furthermore, E297K is a naturally occurring inactivating mutation that can lead to life-threatening hyperparathyroidism (*Bai et al., 1996*; *Hendy et al., 2009*; *Pollak et al., 1993*). These observations suggest that the binding of an amino acid is required for extracellular $Ca^{2+}$-sensing by CaSR.

## $Ca^{2+}$-binding

We identified four distinct $Ca^{2+}$-binding sites within each protomer of the active structure using anomalous difference maps, and named these sites 1 through 4 (or 1'-4' in the second protomer) (*Figure 5A,B*; *Figure 5—figure supplement 1*; *Table 1—source data 1*). The inactive structure revealed electron density at site 2 that is consistent with a bound $Ca^{2+}$ ion (*Figure 5C,D*). None of the $Ca^{2+}$-binding sites observed in the CaSR ECD structures has been reported previously.

Site 1 is located in a loop region at the top of LB1 domain (*Figure 5A,B*). The bound $Ca^{2+}$ ion is primarily coordinated by backbone carbonyl oxygen atoms of I81, S84, L87, and L88 (site 1'). The structural configuration of site 1 is similar in the inactive and active structures even though it is only occupied in the active state (*Figure 5—figure supplement 1*). The disease-causing mutation I81M (*Hendy et al., 2009*) is located at site 1, and it abolished $Ca^{2+}$-dependent receptor response, possibly by disrupting a tightly packed hydrophobic patch adjacent to $Ca^{2+}$-binding site 1 (*Figure 6A,B*).

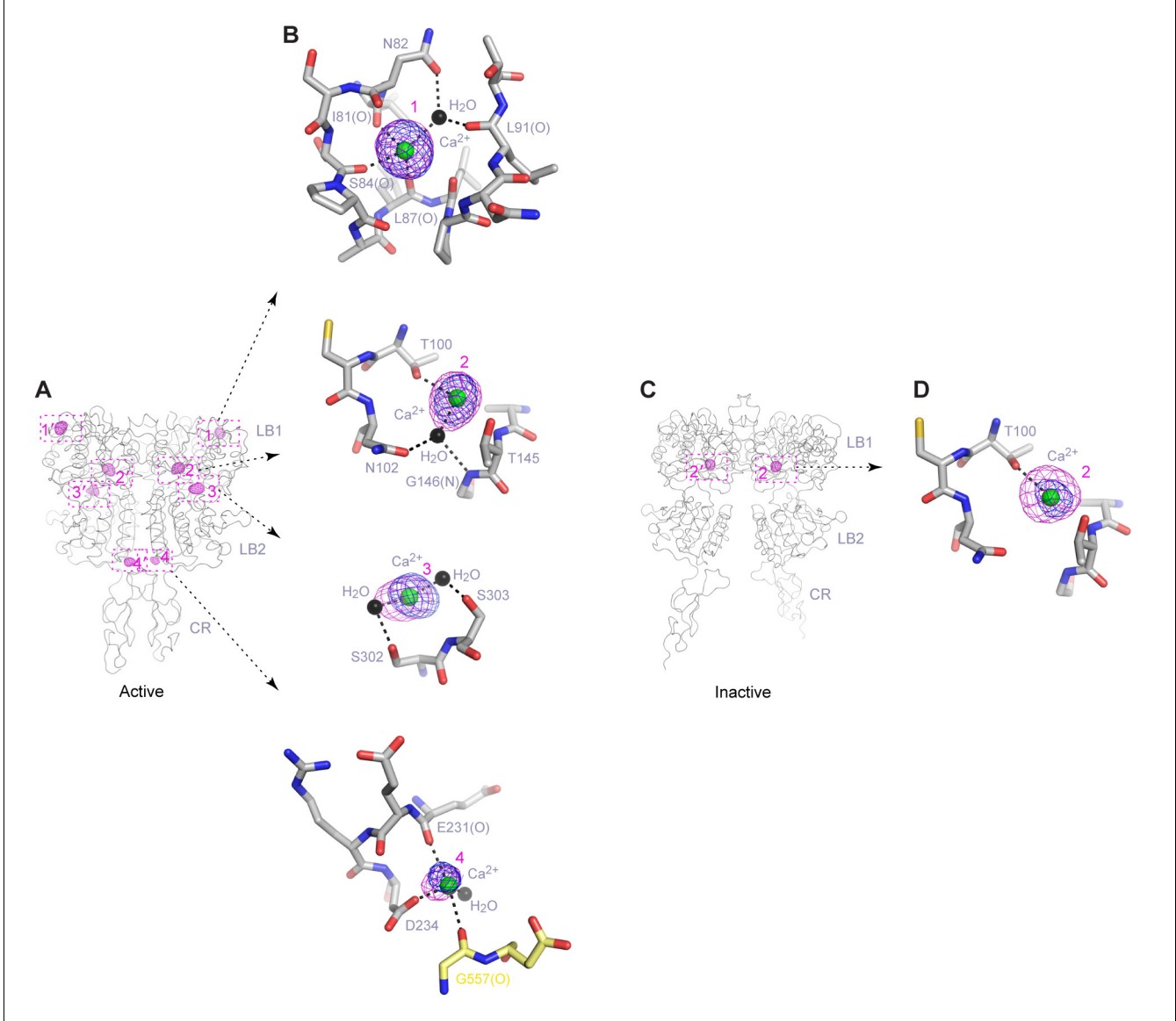

**Figure 5.** Ca$^{2+}$-binding sites. (**A**) Active-state structure showing peaks in anomalous difference Fourier map (magenta mesh; 3σ contour level) that correspond to bound Ca$^{2+}$ ions. Sites are labeled 1–4 or 1'-4' for each protomer. (**B**) Specific contacts between CaSR ECD and each bound Ca$^{2+}$ ion within one protomer of the active structure. Anomalous difference Fourier map (magenta): sites 1–3, 6σ; site 4, 4.5σ. Fo-Fc difference map (blue): sites 1–3, 4.5σ; site 4, 2.5σ. Distances between Ca$^{2+}$ and oxygen atoms (dashed lines) are within 3.0 Å. Dashed lines between water and protein atoms are hydrogen bonds. (**C**) Inactive-state structure showing peaks in anomalous difference Fourier map (magenta mesh; 3σ) at Ca$^{2+}$-binding sites 2 and 2'. (**D**) Specific contacts between CaSR ECD and bound Ca$^{2+}$ ion within one protomer of the inactive structure. Anomalous difference Fourier map (magenta): 5σ. Fo-Fc difference map (blue): 4.5σ.

The following figure supplement is available for figure 5:

**Figure supplement 1.** Ca$^{2+}$-binding sites in the active homodimer.

This implies that the local conformation of this loop region is important for receptor function, and the Ca$^{2+}$ ion stabilizes the observed conformation in the active state.

The inactive and active structures share a common Ca$^{2+}$-binding mode at site 2, suggesting that the bound Ca$^{2+}$ is an integral part of the CaSR structure (*Figure 5A–D*; *Figure 5—figure supplement 1*). Site 2 is positioned directly above the interdomain crevice in LB1 domain, and it abuts the L-Trp binding site in the cleft. The Ca$^{2+}$ ion is coordinated by the hydroxyl group of T100 in both

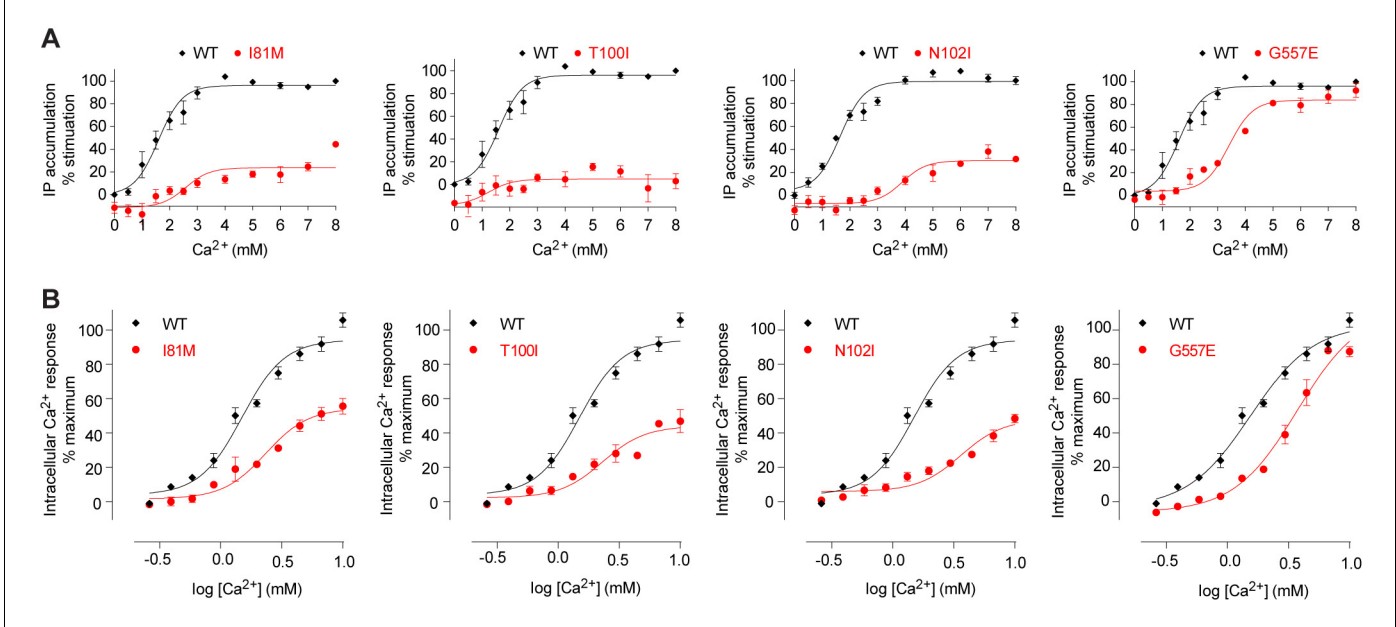

**Figure 6.** Mutational analysis of $Ca^{2+}$-binding sites. (A, B) Dose-dependent $Ca^{2+}$-stimulated IP accumulation (A) and intracellular $Ca^{2+}$ mobilization (B) in cells transiently expressing wt or mutant CaSR. Naturally occurring inactivating mutations I81M and T100I are located at various $Ca^{2+}$-binding sites. The single mutation N102I was designed based on structure to interfere with $Ca^{2+}$-binding.

states, and by the carboxyl group of N102 through a water molecule in the active structure. In addition, T145, another residue lining the site, forms part of the L-Trp binding cleft in the active state. Therefore, an intact $Ca^{2+}$ site 2 provides an essential framework for L-Trp recognition. Indeed, introducing a hydrophobic residue at this site through mutations T100I, N102I, or T145I nearly eliminated $Ca^{2+}$-induced receptor activity (*Figure 4D*; *Figure 6A,B*).

Site 3 is positioned at the edge of the interdomain cleft in LB2 domain (*Figure 5A,B*). The $Ca^{2+}$ ion is coordinated by the hydroxyl groups of S302 and S303 either directly (site 3′) or indirectly through water molecules. Alignment of the inactive and active structures in this region showed a small conformational change (*Figure 5—figure supplement 1*). $Ca^{2+}$ ion stabilizes a loop conformation that permits an interdomain hydrogen bond between the neighboring LB2 residue S301 and LB1 residue R66. Such interaction enhances domain closure of CaSR ECD for receptor activation.

Among all four $Ca^{2+}$-binding sites in CaSR ECD structure, site 4 is most closely associated with receptor activation because it directly participates in the formation of an active receptor conformation. Site 4 is part of the homodimer interface formed upon agonist binding, bridging the LB2 domain of one subunit and CR domain of the second subunit (*Figure 5A,B*). The $Ca^{2+}$ ion is coordinated by three interfacial residues, including the carboxylate group of D234 and carbonyl oxygen of E231 and G557 (*Figure 5B*). The natural mutation G557E (*Hendy et al., 2009*) reduced the potency of $Ca^{2+}$ possibly by affecting backbone conformation, thereby weakening the affinity of $Ca^{2+}$ for this site (*Figure 6A,B*). Our structural data indicate that the $Ca^{2+}$ ion at site 4 stabilizes the active conformation of the receptor by facilitating homodimer interactions between the membrane-proximal LB2 and CR domains.

The $Ca^{2+}$ ions have different peak heights in the anomalous difference maps, which are correlated with different $Ca^{2+}$-occupancies at various sites. The $Ca^{2+}$ ions at sites 1 and 2 have strong peaks (12.1–13.1 σ), indicating that these are high-occupancy sites. The $Ca^{2+}$ ions at sites 3 (7.3 – 9.0 σ) and 4 (5.8 σ) have weaker anomalous peaks, which suggest low occupancy and possibly low affinity. The $Ca^{2+}$ ion at site 4 has the weakest peak compared with other sites, which is consistent with the site being occupied only at elevated $Ca^{2+}$ concentration for receptor activation.

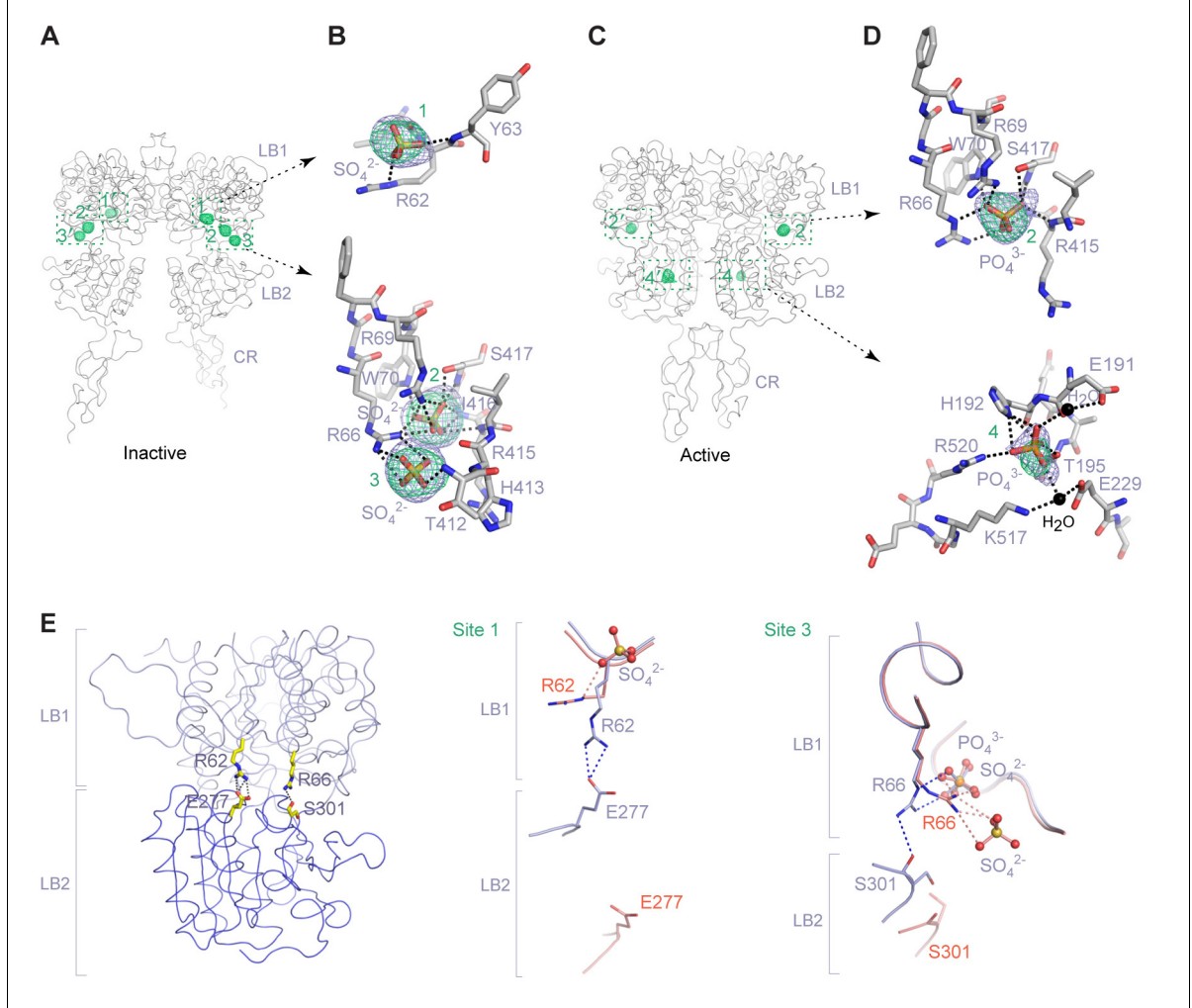

**Figure 7.** Anion-binding sites. (**A**) Inactive-state structure showing peaks in anomalous difference Fourier map (green mesh; 3σ) that correspond to bound $SO_4^{2-}$ ions. Sites are labeled 1–3 or 1'-3' for each protomer. (**B**) Specific contacts between CaSR ECD and each bound $SO_4^{2-}$ ion within one protomer of the inactive structure. Anomalous difference Fourier map (green): 3.5σ. Fo-Fc map (blue): 4σ. Dashed lines represent hydrogen bonds. (**C**) Active-state structure showing peaks in anomalous difference Fourier map (green mesh; 3σ) that correspond to bound $PO_4^{3-}$ ions. Sites are labeled 2 and 4 or 2' and 4' for each protomer. (**D**) Specific contacts between CaSR ECD and each bound $PO_4^{3-}$ ion within one protomer of the active structure. Anomalous difference Fourier map (green): 3.5σ. Fo-Fc map (blue): 4σ. (**E**) Active-state structure of CaSR ECD showing the additional hydrogen bonds formed across the interdomain cleft in the absence of any bound anion at sites 1 and 3 (left). Comparison of inactive (red) and active (light blue) structures in the region of anion binding sites 1 (center) and 3 (right).

The following figure supplement is available for figure 7:

**Figure supplement 1.** Anion-binding sites in the active homodimer.

## Anion binding

We identified a total of four anion-binding sites in the inactive and active CaSR ECD structures based on anomalous difference maps (1–4 or 1'-4' in the second protomer) (*Figure 7*; *Figure 7—figure supplement 1*; *Table 1—source data 1*). Sites 1–3 are located above the interdomain cleft in the LB1 domain, and site 4 is part of LB2 domain.

In the inactive structure, electron densities revealed that anions were bound at sites 1–3 (*Figure 7A,B*). In the active structure, only sites 2 and 4 are occupied (*Figure 7C,D*). We modeled the anions as $SO_4^{2-}$ ions in the inactive state and $PO_4^{3-}$ ions in the active state given their respective

presence in the crystallization reagents. It is also possible that endogenous anions other than $SO_4^{2-}$ and $PO_4^{3-}$ are bound at these sites.

The anion at site 1 is coordinated by the guanidine group of R62 and backbone nitrogen of Y63 (*Figure 7B*). In the absence of a bound anion in the active state, the side chain of LB1 residue R62 reaches across the interdomain cleft to form a salt bridge with E277 of LB2 domain (*Figure 7E*). This contact stabilizes the closed conformation of the CaSR VFT module.

The anion bound at site 2 is held in place by multiple hydrogen bonds with the side chains of R66, R69, W70, and S417, and main chains of R415, I416, and S417 (*Figure 7B,D*). The structural integrity of this site is important for receptor function. Each of the mutations R66H, R69E, and S417L essentially eradicated receptor signaling (*Figure 8A,B*). Among these, R66H is a disease-associated mutation (*Hendy et al., 2009*; *Pidasheva et al., 2006*).

Anion-binding site 3 is adjacent to site 2. The bound anion is also coordinated by R66 and additionally by T412 (*Figure 7B*). In the active state, the side chain of LB1 residue R66 forms a hydrogen bond with S301 of LB2 domain (*Figure 7E*). This interaction also serves to maintain the CaSR ECD in a closed conformation.

The anion at site 4 is coordinated by H192, T195, K225 and R520 in the active structure (*Figure 7D*). In the absence of a bound anion, the side chains of several binding-site residues are disordered in the inactive structure. This indicates that the anion serves to stabilize the local conformation of the receptor structure.

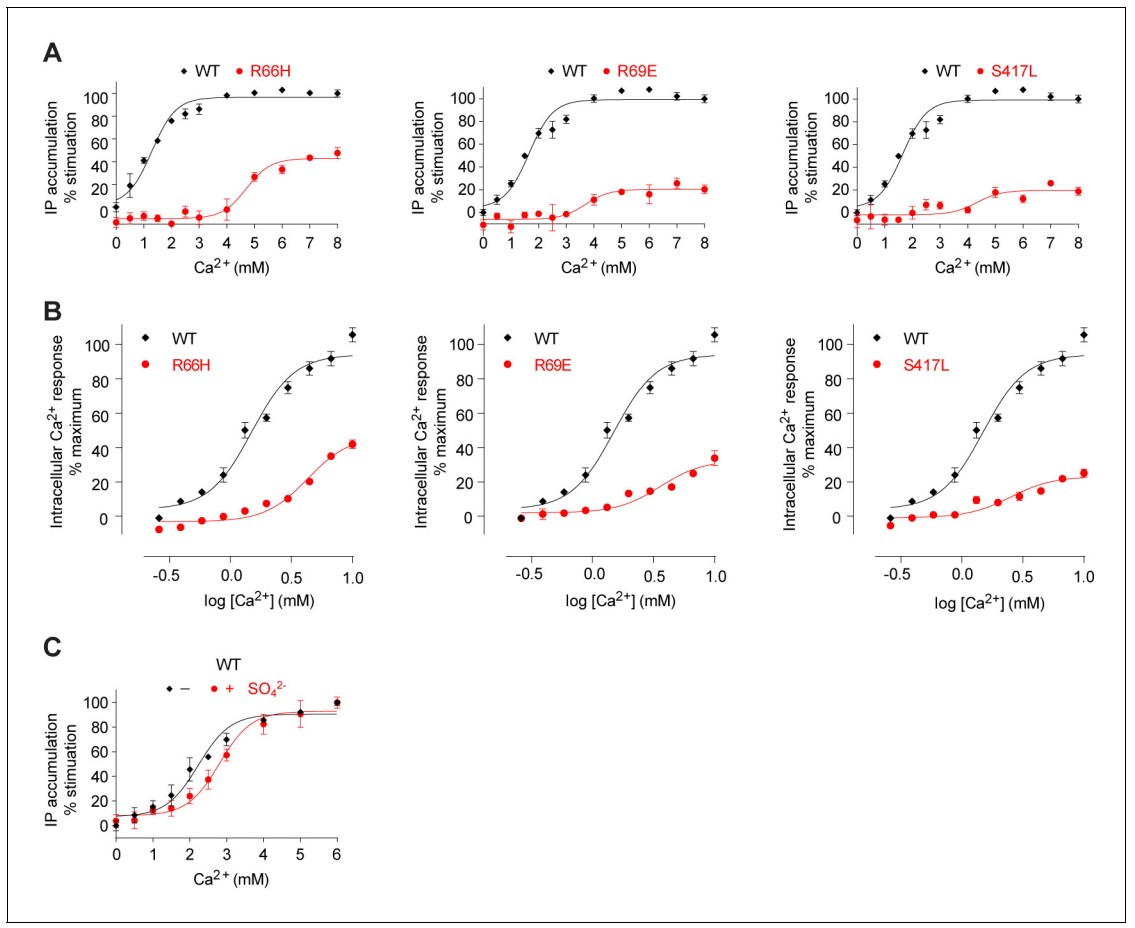

**Figure 8.** Mutational analysis of anion-binding sites. (A, B) Dose-dependent $Ca^{2+}$-stimulated IP accumulation (A) and intracellular $Ca^{2+}$ mobilization (B) in cells transiently expressing wt or mutant CaSR. Naturally-occurring inactivating mutations R66H, R69E, and S417L are located at anion-binding site 2 (or 2'). (C) Effect of $SO_4^{2-}$ ion on $Ca^{2+}$-stimulated IP accumulation in cells transiently expressing wild-type CaSR.

We measured the effect of anion on $Ca^{2+}$-dependent receptor response and found that the presence of $SO_4^{2-}$ slightly decreased receptor activity, increasing the $EC_{50}$ of $Ca^{2+}$ by approximately 25% (*Figure 8C*). This finding confirmed that anions have a negative allosteric effect on the receptor.

## Discussion

### Implications for agonist-dependent receptor activation

Our structural analyses of CaSR ECD provide direct evidence that amino acids are agonists of CaSR, and they act concertedly with $Ca^{2+}$ to achieve full receptor activation. L-Trp, the amino acid used in this study, fits the role of an orthosteric agonist for CaSR. (1) It binds at the interdomain crevice of the VFT module, the canonical agonist-binding site for class C GPCRs (*Geng et al., 2013*; *Kunishima et al., 2000*; *Muto et al., 2007*; *Tsuchiya et al., 2002*). (2) L-Trp shares a common receptor-binding mode with the endogenous agonists of mGluRs and GABA$_B$ receptor, which are also amino acids or their analogs (*Geng et al., 2013*; *Kunishima et al., 2000*; *Muto et al., 2007*; *Tsuchiya et al., 2002*). The residues involved in agonist recognition are located at the same positions in the structures of these receptors (*Figure 4—figure supplement 1*). For example, a conserved serine residue is responsible for securing the carboxylate of L-Trp, glutamate and GABA in CaSR (S147), mGluRs (S165) (*Kunishima et al., 2000*) and GABA$_B$ receptor (S130) (*Geng et al., 2013*), respectively. (3) L-Trp interacts with both LB1 and LB2 domains to facilitate extracellular domain closure, a crucial first step during CaSR activation. In contrast, no $Ca^{2+}$ ion is found at the putative orthosteric agonist-binding site to induce domain closure. (4) Mutations of L-Trp-binding residues completely blocked $Ca^{2+}$-induced IP accumulation and intracellular $Ca^{2+}$ mobilization (*Silve et al., 2005*; *Zhang et al., 2002*), indicating that L-Trp is required for $Ca^{2+}$-mediated receptor response. (6) L-Trp directly activates CaSR-mediated intracellular $Ca^{2+}$ mobilization in the presence of extracellular $Ca^{2+}$.

On the other hand, $Ca^{2+}$ ion fulfills at least three functional roles. First, it maintains the structural integrity of CaSR, as manifested by the importance of $Ca^{2+}$-binding site 2 for receptor function. Second, it is directly involved in receptor activation. Specifically, the $Ca^{2+}$ ion at site 4 stabilizes the unique homodimer interface between membrane-proximal LB2 and CR domains in the active state. Third, $Ca^{2+}$ enhances L-Trp binding to CaSR ECD, possibly by reinforcing the L-Trp bound and active conformation of the receptor. Furthermore, the actions of $Ca^{2+}$ and amino acids on CaSR are interdependent. While the presence of extracellular $Ca^{2+}$ above a threshold level is required for amino-acid-mediated CaSR activation, amino acids increase the sensitivity of the receptor toward $Ca^{2+}$. Taken together, we conclude that amino acids and $Ca^{2+}$ ions are indeed co-agonists of CaSR, acting jointly to trigger receptor activation.

This then led us to an intriguing question: How does $Ca^{2+}$ ion activate the receptor on its own as demonstrated in various cell-based functional assays? It is possible that amino acids are present in cell culture media and extracellular fluid at sufficient concentration to prime the receptor to respond to increasing concentrations of $Ca^{2+}$. We have obtained crystals of CaSR ECD in the absence of any additional amino acids. Nevertheless, the structure showed a stretch of continuous density in the interdomain cleft that may belong to an endogenous ligand (*Figure 4—figure supplement 2*; *Table 1—source data 2*). Despite multiple attempts, we have not been able to identify the structure of this endogenous ligand through mass spectrometry. In light of the L-Trp-bound CaSR ECD structure, we reason that the endogenous ligand may be an amino acid or even a mixture of amino acids. In the inactive structure, the endogenous ligand was likely removed by citrate buffer (pH 5.5) during enzymatic deglycosylation.

Metabolic balances of $Ca^{2+}$ and $PO_4^{3-}$ are linked through hormonal factors such as parathyroid hormone and fibroblast growth factor-23, which control the homeostasis of both ions (*Brown, 2013*; *Quinn et al., 2013*; *Tyler Miller, 2013*). We therefore reason that the physiologically relevant anion bound to CaSR is likely $PO_4^{3-}$, which primarily exists in a $HPO_4^{2-}/H_2PO_4^-$ mixture in biological systems. First, the anion at site 2 is required for structural stability of the receptor. Second, the anions at sites 1 and 3 appear to stabilize the inactive conformation. Binding of anions at these sites prevent favorable interactions across the interdomain cleft that promote VFT closure. Third, the presence of anion decreases CaSR-mediated IP accumulation. Therefore, these anions may exert a negative modulatory effect on CaSR activity.

The presence of anion-binding sites in CaSR may also provide a mechanism for CaSR to sense polycations such as polyamines. Increasing concentrations of polyamines could potentially compete with arginine residues at the anion-binding sites to bind to $PO_4^{3-}$, thereby prompting the dissociation of $PO_4^{3-}$ from relatively weak sites, and releasing their inhibitory effect on the receptor. This would drive CaSR toward its active-state conformation. Our hypothesis would predict that polycations with higher number of positive charges will be more effective agonists. Indeed, previous studies have shown that polyamines mediate an increase in intracellular inositol phosphate and $Ca^{2+}$ accumulation with the rank order seprmine > spermidine > putrescine (*Cheng et al., 2004*; *Quinn et al., 1997*).

In summary, activation of CaSR involves an intricate interplay of amino acids, $Ca^{2+}$, and possibly $PO_4^{3-}$ ions. Like other GPCRs (*Rosenbaum et al., 2011*), CaSR exists in a conformational equilibrium between inactive and active states (*Figure 9*). (1) CaSR adopts an open conformation in the resting state, and $PO_4^{3-}$ ions promote the inactive configuration. (2) An L-amino acid closes the groove in the extracellular VFT module, thereby inducing the formation of a novel homodimer interface between subunits. (3) $Ca^{2+}$ ions stabilize the active state by enhancing homodimer interactions between membrane-proximal domains to fully activate the receptor. The combination of agonist-induced VFT closure and specific association of membrane-proximal CR domains in CaSR will likely lead to rearrangement of the transmembrane domains for receptor activation.

## Common class C GPCR activation mechanism

Structural and functional data suggest a universal activation mechanism for class C GPCRs. First, agonist causes VFT closure in all three receptor systems of CaSR, mGluRs and GABA_B receptor. Second, receptor activation requires the association of membrane-proximal domains. For CaSR, this involves the formation of a novel homodimer interface between the LB2 and CR domains. For GABA_B receptor, which lacks the CR region, agonist leads to the formation of a large heterodimer interface between the LB2 domains (*Geng et al., 2013*). Similarly for mGluRs, single-molecule fluorescence resonance energy transfer (FRET) studies indicate that the LB2 domains of mGluRs come into proximity to stabilize the active state (*Vafabakhsh et al., 2015*). Furthermore, disulfide crosslinking experiments of mGluR demonstrate that a precise association between the CR domains is sufficient for full receptor activation (*Huang et al., 2011*). Third, agonist binding is accompanied by a decrease in separation between the C-terminal ends of extracellular domains. This will likely result in rearrangement of the transmembrane domain dimer for receptor activation. Indeed, FRET and crosslinking studies have detected movement between interacting transmembrane domains of mGluRs and GABA_B receptor upon activation (*Matsushita et al., 2010*; *Tateyama et al., 2004*; *Xue et al., 2015*). In conclusion, agonist-induced VFT closure that leads to the specific association of membrane-proximal domains is a common mechanism shared by all class C GPCRs during ligand-dependent receptor activation.

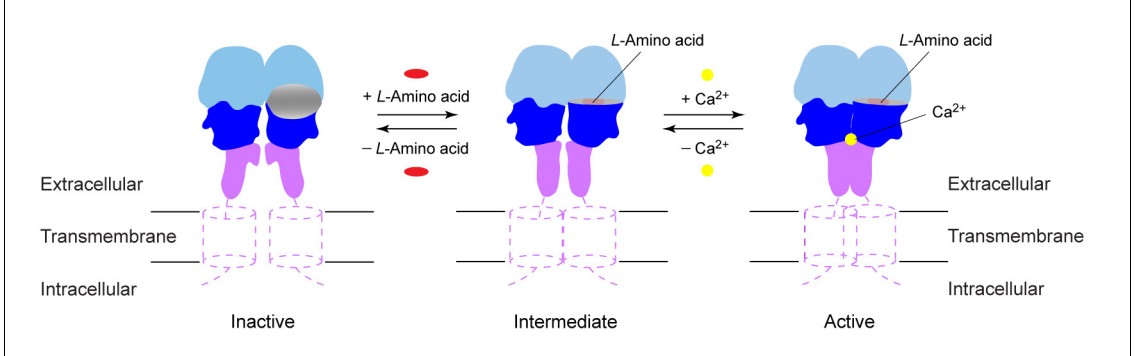

**Figure 9.** Activation mechanism of CaSR. The schematic diagram shows the equilibrium between the resting and active states of CaSR and the effects of L-amino acid and $Ca^{2+}$ binding.

## Materials and methods

### Protein expression and purification

The extracellular domain of human CaSR (1–612) was cloned into the pFBDM vector (*Berger et al., 2004*) for expression in baculovirus-infected insect cells. Wild-type CaSR was heavily glycosylated, and we took two approaches to remove carbohydrates from CaSR ECD: (1) elimination of potential glycosylation sites through mutation, and (2) enzymatic deglycosylation.

Mutants of CaSR ECD were generated in which either two or three N-linked glycosylation sites were eliminated. Both mutant constructs contained the mutations N386Q and S402N, while one also had the additional mutation N468Q. A Flag tag was engineered at the C-terminus of each construct to facilitate affinity purification. In agreement with previous studies of full-length CaSR in mammalian cells (*Ray et al., 1998*), we found that expression of CaSR ECD in insect cells was essentially eliminated when more than three glycosylation sites were disrupted. The CaSR ECD glycosylation mutants were constructed to maximize the reduction of carbohydrate content while maintaining a sufficient expression level for structural studies.

Wild-type and mutant CaSR ECD were secreted from sf9 insect cells infected with the corresponding recombinant CaSR ECD virus. The CaSR ECD protein was isolated from cell supernatant by anti-Flag antibody (M2) affinity chromatography, and eluted with 100 μg/ml Flag peptide in 50 mM Tris, pH 7.5 and 150 mM NaCl. The protein was further purified by gel filtration chromatography (superdex 200, GE Healthcare Life Sciences, USA) in 20 mM Tris, pH 8.0 and 150 mM NaCl to remove aggregates. Finally, the CaSR ECD protein was applied to an ion exchange column (MonoQ, GE) in 20 mM Tris, pH 8.0 and eluted using a linear salt gradient from 0 to 1 M NaCl. All purification procedures were performed in the presence of 10 mM $CaCl_2$. To isolate the L-Trp-bound receptor, purified CaSR ECD protein was washed extensively with a solution containing 20 mM Tris, pH 8.0, 10 mM L-Trp, and 10 mM $CaCl_2$ prior to crystallization. This would facilitate the complete saturation of L-Trp- and $Ca^{2+}$-binding sites within the receptor.

We applied an enzymatic deglycosylation procedure to wild-type receptor and the CaSR ECD mutant with two glycosylation-site mutations. Each construct was expressed in sf9 cells in the presence of the N-glycosylation processing inhibitor kifunensine (1 mg/L) to produce high mannose glycoproteins that were sensitive to enzymatic cleavage by endoglycosidase H (Endo H). The cell supernatant was applied to an M2 anti-Flag antibody affinity column, and the bound CaSR ECD protein was eluted with 100 μg/ml Flag peptide in 50 mM Tris, pH 7.5 and 150 mM NaCl. Well-folded CaSR ECD was separated from aggregates by gel filtration chromatography in 20 mM Tris, pH 8.0, and 150 mM NaCl. The CaSR ECD protein was subsequently digested overnight with Endo H in 50 mM Na Citrate, pH 5.5. The partially deglycosylated CaSR ECD protein was further purified by ion exchange chromatography in 20 mM Tris, pH8.0 using a linear salt gradient from 0 to 1 M NaCl. The protein purification and enzymatic deglycosylation procedures were performed either in the absence or presence of 2 mM $CaCl_2$, which is within the normal range of $Ca^{2+}$ concentrations in plasma.

### Crystallization and data collection

Wild-type CaSR ECD did not form well-diffracting crystals possibly because the presence of flexible and heterogeneous carbohydrates on the protein surface interfered with crystallization.

The CaSR ECD mutant carrying two glycosylation-site mutations formed diffracting crystals after enzymatic deglycosylation. The crystals were obtained at 20°C in 1.5 M $Li_2SO_4$, 100 mM Tris pH 8.5 in the absence and presence of 2 mM $CaCl_2$. Protein crystallization was achieved by hanging drop vapor diffusion method using 24-well VDX plates (Hampton Research, USA). The CaSR ECD crystals were flash-cooled with liquid nitrogen in a cryoprotecting solution containing 3.0 M $Li_2SO_4$, 100 mM Tris pH 8.5, and the same concentration of $CaCl_2$ as used for crystallization. These CaSR ECD crystals have the space group F222 with one molecule per asymmetric unit (form I).

The CaSR ECD mutant carrying three glycosylation-site mutations crystallized under two different conditions. (1) In the absence of additional amino acids, CaSR ECD formed the best diffracting crystals at 20°C in 0.9 M Na Citrate, 100 mM Tris, pH 8.0, and 10 mM $CaCl_2$. The crystals were obtained using hanging drop vapor diffusion technique. These crystals were flash-cooled with liquid nitrogen in a cryoprotecting solution containing 0.9 M Na Citrate, 100 mM Tris, pH 8.0, 10 mM $CaCl_2$, and

20% glycerol. (2) Crystals of the L-Trp-bound CaSR ECD mutant were grown at 20°C in 1.6 M $NaH_2PO_4$, 0.4 M $K_2HPO_4$, 100 mM $Na_2HPO_4$/citric acid pH 4.2, 10 mM $CaCl_2$, and 10 mM L-Trp. The crystals were obtained by sitting drop vapor diffusion method using 24-well Cryschem plates (Hampton Research). These crystals were flashed-cooled with liquid nitrogen in a cryoprotecting solution containing 20% glycerol and all other components of the crystallization solution. The crystals obtained from both conditions belong to space group C2 and have two molecules per asymmetric unit (form II).

Diffraction data were collected at the 24ID-C and 24ID-E beamlines of Advanced Photon Source (APS). To minimize radiation damage at low energy, a native dataset was collected at the energy of Se-K edge (λ=0.9792 Å) for a single form I crystal of CaSR ECD grown in the presence of 2 mM $Ca^{2+}$. This native dataset was used to solve the inactive-state CaSR ECD structure. Form I crystals grown in the absence of additional $Ca^{2+}$ diffracted to much lower resolution than those obtained in the presence of 2 mM $Ca^{2+}$ and were therefore not used for structural determination.

Multiple anomalous data sets were also collected at λ = 1.7712 Å for form I crystals of CaSR ECD, each from a single crystal. A total of five anomalous data sets were obtained for CaSR ECD in the absence of $Ca^{2+}$. These data sets were integrated individually by XDS (*Kabsch, 2010*), and then scaled and merged using CCP4 programs (*Winn et al., 2011*) to calculate the anomalous difference Fourier maps in the absence of additional $Ca^{2+}$. Similarly, eight anomalous data sets were collected for CaSR ECD in the presence of 2 mM $Ca^{2+}$, and combined to determine the anomalous difference Fourier maps in the presence of plasma concentration of $Ca^{2+}$. The anomalous difference maps obtained in the absence and presence of $Ca^{2+}$ revealed peaks that correspond to bound $Ca^{2+}$ and $SO_4^{2-}$ ions at the same positions in the CaSR ECD structure. The averaging of multiple data sets enhanced signal to noise in anomalous diffraction (*Liu et al., 2014*).

A total of four anomalous data sets were collected at low energy (λ=1.7712 Å) for L-Trp-bound form II crystals of CaSR ECD, each from a single crystal. The four data sets were processed individually by XDS (*Kabsch, 2010*) and CCP4 programs (*Winn et al., 2011*). The data set with the highest resolution limit (2.6 Å) was used to determine the active-state CaSR ECD structure in the presence of L-Trp and excess $Ca^{2+}$. The four data sets were also scaled and merged for the calculation of anomalous difference Fourier maps.

## Structure determination

The structure of L-Trp-bound CaSR ECD in form II crystal was solved by molecular replacement. A two-fold non-crystallographic symmetry (NCS) axis was identified from the self-rotation function. Polyalanine models generated from the individual LB1 and LB2 domains of mGluR3 ECD structure (*Muto et al., 2007*) (PDB code: 2E4U) were used as search probes to locate the VFT modules of both CaSR ECD molecules in the crystal. After phase improvement by two-fold NCS-averaging, additional density appeared for the CR domain of CaSR ECD. A complete model of the CaSR ECD homodimer was developed through a succession of manual fittings and iterative refinement. The final model contained the CaSR ECD residues 20–119, 135–359 and 393–598 in one protomer, and residues 22–122, 136–360 and 392–602 in the other protomer. Each protomer contained eight intrasubunit disulfide bridges. Electron density was visible for carbohydrate residues (N-acetyl-glucosamine) attached to Asn90, Asn287, Asn488, Asn541 of one protomer, and Asn541 of its dimer partner. Finally, each protomer was also bound to one L-Trp ligand, four $Ca^{2+}$ ions, and two $PO_4^{3-}$ ions. The $Ca^{2+}$ and $PO_4^{3-}$ ions were identified by anomalous difference Fourier maps calculated using data collected at a wavelength of 1.7712 Å. We modeled $PO_4^{3-}$ as the anions in the CaSR ECD structure because they were major components of the crystallization solution. The anomalous scattering of $Ca^{2+}$ at 1.75 Å has been used successfully to identify the $Ca^{2+}$-binding sites in a voltage-gated calcium channel (*Tang et al., 2014*). Ramachandran analysis places 94.9% of all residues in favored regions and 0.28% in outlier regions.

A molecular replacement solution was also obtained for CaSR ECD in form II crystal in the presence of 10 mM $Ca^{2+}$ but absence of any additional amino acid. The protein structure of L-Trp-bound CaSR ECD without any bound ligand was used as the search model. A stretch of electron density at the interdomain cleft region indicates the presence of an endogenous ligand. This structure was partially refined because we have not been able to identify the structure of the endogenous ligand by mass spectrometry despite multiple attempts.

The structure of partially deglycosylated CaSR ECD in form I crystal was solved using the individual LB1, LB2, and CR domains of L-Trp-bound CaSR ECD as search models. The asymmetric unit of the form II crystals contained one CaSR ECD protomer, and it forms a disulfide-linked homodimer with a crystallographic symmetry-related molecule. The final model for a single protomer contained CaSR ECD residues 21–130 and 136–603. In addition to the eight intrasubunit disulfide linkages within each protomer, the inter-subunit disulfide bond formed by C129 was ordered in the form I CaSR ECD crystal structure. Electron density was also visible for carbohydrate residues (N-acetyl-glucosamine) attached to Asn261, Asn287, Asn446, Asn468, Asn488, Asn541, and Asn594. In addition, each protomer had one $Ca^{2+}$ ion, and three $SO_4^{2-}$ ions. The $Ca^{2+}$ and $SO_4^{2-}$ ions were identified by anomalous difference Fourier maps calculated from data collected at 1.7712 Å. We reasoned that $SO_4^{2-}$ ions were most likely the anions bound to CaSR ECD since they were used as the precipitant for crystallization. Geometric analysis places 92.9% of all residues in favored regions and 0.35% as outliers. In the form I CaSR ECD structure, the interdomain groove was empty except for a water molecule; it also forms crystal contacts with a symmetry-related molecule.

Molecular replacement searches were carried out using PHASER (*McCoy et al., 2007*). Model building was performed with COOT (*Emsley and Cowtan, 2004*). Structural refinement was executed using BUSTER (*Roversi et al., 2000*). Anomalous difference Fourier maps were calculated with PHENIX (*Adams et al., 2010*). Ramachandran statistics were obtained for each structure using MolProbity (*Chen et al., 2010*). Pairwise structural comparison was performed using LSQMAN (*Novotny et al., 2004*). Protein contacts were analyzed using the CCP4 program CONTACT (*Collaborative Computational Project, 1994*). Software installation support was provided by SBGrid (*Morin et al., 2013*).

## Cell surface expression

Full-length human CaSR was cloned into a pcDNA3.1(+) vector (Life Technologies, USA) for expression in human embryonic kidney (HEK) 293 cells. A Flag tag was inserted after the signal peptide of CaSR. Mutants of CaSR were constructed using the QuikChange mutagenesis system (Agilent Technologies, California, USA).

HEK293 T/17 cells (ATCC) were transfected by Lipofectamine 2000 (Life Technologies) with wild-type or mutant CaSR plasmids. Cells permeabilized with 0.5% Triton X100 were used to determine the total expression level of CaSR in transfected cells. Untreated cells were used to determine the cell surface expression level of the receptor. The amount of surface protein detected for each construct was normalized to that found in the total cell lysate.

The cells were blocked with 5% milk, and then incubated with mouse anti-Flag M1 antibody (Sigma-Aldrich, USA) as the primary antibody to measure CaSR expression. Donkey anti-mouse IRDye 800-labeled antibody (LiCor Biosciences, Nebraska, USA) was used as the secondary antibody. Fluorescent signals were measured with an Odyssey Infrared Imager (LiCor). The results of three independent experiments were used for statistical analysis.

Most of the mutants were expressed on the cell surface at levels comparable to that of the wild-type receptor. The exceptions were R66H, R69E, I81M, T100I, N102I, and S417L, which reduced the surface expression of the mutant receptors to approximately 70–75% of the wild-type level.

## Inositol phosphate measurement

Measurement of inositol phosphate (IP) accumulation was carried out using the homogenous time-resolved fluorescence (HTRF) IP-one Tb kit (Cisbio Bioassays, USA). This assay quantifies the accumulation of inositol 1-monophosphate ($IP_1$), a degradation product of inositol 1,4,5-triphosphate ($IP_3$) that is stable in the presence of LiCl. Briefly, HEK293 T/17 cells were transiently transfected with wild-type or mutant full-length CaSR plasmids. The cells were stimulated with increasing concentrations of $Ca^{2+}$ two days post transfection in a buffer containing 10 mM HEPES pH 7.4, 0.5 mM $MgCl_2$, 4.2 mM KCl, 146 mM NaCl, 5.5 mM glucose, and 50 mM LiCl. The reaction mixture was then incubated with an IP1 analog coupled to a d2 fluorophore (acceptor) and an anti-IP1 monoclonal antibody labeled with Eu Cryptate (donor). The IP1 produced by cells upon activation of CaSR competes with IP1 coupled to the dye d2 for binding to the anti-IP1 antibody. The resulting FRET signal is inversely proportional to the concentration of IP1 in the sample. The fluorescence data was acquired at 620 and 665 nm using an EnVision plate reader (Perkin Elmer, USA) after laser excitation

at 320 nm. The FRET signal was calculated as the fluorescence ratio (665 nm/620 nm). Basal activity was determined in the absence of $Ca^{2+}$ stimulation. The percent stimulation of each receptor mutant was calculated based on the wild-type response obtained under the same condition. Data analysis was performed using the non-linear regression algorithms in Prism (GraphPad Software, USA). Data points represent average ± s.e.m. of triplicate measurements.

The effect of anion on $Ca^{2+}$-stimulated IP accumulation was determined using $SO_4^{2-}$ instead of $PO_4^{3-}$ because of the modest solubility of $CaHPO_4$ (1.5 mM), the predominant form of calcium phosphate salt at physiological pH (7.4). Specifically, dose-dependent $Ca^{2+}$-induced IP accumulation was measured in the absence and presence of 10 mM $Li_2SO_4$ in the reaction buffer.

## Intracellular $Ca^{2+}$ flux measurement

Measurement of intracellular $Ca^{2+}$ mobilization was performed using a FLIPR Fluorescent Imaging Plate Reader (FLIPR) Calcium Assay kit (Molecular Devices). Briefly, HEK293 cells were transiently transfected with wild-type or mutant full-length CaSR plasmids and cultured overnight. The cells were incubated in a loading medium containing 50% Opti-MEM, 50% $Ca^{2+}$-free Hank's balanced salt solution, 2.5% fetal bovine serum, 20 mM HEPES pH 7.4, 2.5 mM probenecid and 2 μM fluorescent $Ca^{2+}$ indicator Fluo-4 AM (Life Technologies) for 1 hr, and then placed into the FLIPR. $CaCl_2$ (prepared in Hank's balanced salt solution and 20 mM HEPES, pH 7.4) was added at 10 s, and changes in fluorescence were monitored over a period of 250 s following excitation at a wavelength of 488 nm and detection at 510–560 nm. Data analysis was performed using the non-linear regression algorithms in Prism (GraphPad Software). Data points represent average ± s.e.m. of triplicate measurements.

To measure the potentiating effect of L-Trp on the response of CaSR to extracellular $Ca^{2+}$, HEK293 cells were transiently transfected with wild-type CaSR plasmid and cultured for 48 hr. The cells were washed three times with an assay buffer (20 mM HEPES, pH 7.4, 1 mM $CaCl_2$, 1 mM $MgCl_2$, 1 mg/ml BSA, 5.5 mM D-glucose, 5.3 mM KCl, 138 mM NaCl, 4.2 mM $NaHCO_3$, 0.44 mM $KH_2PO_4$, 0.34 mM $Na_2HPO_4$) to remove any endogenously bound ligand, and pre-incubated with 10 mM L-Trp for 20 min before stimulation with extracellular $Ca^{2+}$.

## Single-cell intracellular $Ca^{2+}$ microfluorimetry

HEK293 cells that stably expressed the CaSR (HEK-CaSR cells) were cultured on coverslips in 24-well plates, and loaded in the dark with fura2-AM (5 μM) in physiological saline solution (PSS; 125 mM NaCl, 4 mM KCl, 0.1% w/v D-glucose, 1 mM $MgCl_2$, 20 mM HEPES-NaOH, pH 7.45) that contained 1 mM $CaCl_2$, 0.8 mM $NaH_2PO_4$ and 1 mg/ml BSA for 1.5 hr at 37°C. The cells were washed and stored in a fura2-AM-free loading solution prior to experiments. Fura2-loaded HEK-CaSR cells were transferred into a perifusion chamber, placed in the light path of a Zeiss Axiovert fluorescence microscope (Zeiss, USA), and perifused with PSS containing various concentrations of $Ca^{2+}$ and L-Trp.

Fura2-loaded HEK-CaSR cells were excited by a Lambda DG-4 150 Watt xenon light source (Sutter, Novato, USA), using alternating wavelengths of 340 and 380 nm at 0.5 s intervals, and imaged at 510 nm. For each data set, regions of interest corresponding to the locations of 10 individual cells were selected and digital images were captured using an AxioCam camera controlled by Stallion SB.4.1.0 PC software (Intelligent Imaging Innovations, USA).

Single-cell intracellular $Ca^{2+}$ mobilization data consisted of excitation ratios (F340/F380) plotted against time (min). Ratio data were integrated, expressed as integrated response units (IRUs), and corrected for baseline (PSS containing 0.5 mM $Ca^{2+}$). Concentration-dependent response data were fitted to the equation:

$$R = b + (a - b) C^n / (e^n + C^n)$$

in which: *a* = maximum response; *b* = basal response; *C* = activator concentration; *e* = $EC_{50}$ in mM; and *n* = Hill co-efficient. Estimates of curve-fitting parameters were obtained using Prism (GraphPad Software).

## Scintillation proximity assay

Wild-type CaSR ECD protein was purified by anti-Flag M2 antibody affinity chromatography and gel filtration chromatography in the absence of additional $CaCl_2$ and amino acids. The protein was

incubated in 100 mM NaCitrate, pH 5.5 overnight to remove any endogenously bound ligand. The CaSR ECD protein was then separated from any free ligand by gel filtration chromatography in 20 mM HEPES, pH 8.0 and 150 mM NaCl.

Binding of 0.1 mM L-[$^3$H]Trp (1 Ci/mmol) to 600 ng of purified CaSR ECD was measured with the scintillation proximity assay (SPA) using 1.25 mg/mL Yttrium silicate (YSi) protein A SPA beads (PerkinElmer, USA) in conjunction with 0.125 ng/mL of anti-Flag M2 antibody in 20 mM HEPES, pH 8.0 and 150 mM NaCl at 4°C. Increasing concentrations of non-radioactive L-Trp were added to compete for receptor binding in the presence of 0 mM or 2 mM CaCl$_2$. Each reaction was also performed in the presence of 60 mM non-radioactive L-Trp for background correction. The reactions were allowed to proceed for 1 hr to reach equilibrium. The plates were then counted in a Microbeta counter (PerkinElmer, USA). Data were analyzed using the non-linear regression algorithms in Prism (GraphPad).

## Acknowledgements

We thank WA Hendrickson and RS Kass for advice and support, Q Liu and O Clarke for advice on diffraction data analysis, C Karan and R Realubit for access to EnVision plate reader, K Rajashankar and Northeastern Collaborative Access Team (NE-CAT) staff for help with data collection. The beamlines at NE-CAT are funded by National Institute of Health (NIH) grants P41 GM103403 and S10 RR029205. This work was supported by American Heart Association grant 15GRNT25420002, and NIH grant R01GM112973 (both to QRF). QRF is an Irma Hirschl Career Scientist, Pew Scholar, McKnight Scholar and Schaefer Scholar.

## Additional information

### Funding

| Funder | Grant reference number | Author |
| --- | --- | --- |
| National Institute of General Medical Sciences | R01GM112973 | Qing R Fan |
| American Heart Association | 15GRNT25420002 | Qing R Fan |

The funders had no role in study design, data collection and interpretation, or the decision to submit the work for publication.

### Author contributions

YG, Conceived the study, Designed experiments, Performed protein expression, purification and crystallization, Collected data, Analyzed structures, Measured IP accumulation, Completed molecular biology experiments, Wrote the paper, Revised the paper; LM, Designed experiments, Solved structures, Analyzed structures, Revised the paper; IK, Collected data, Solved structures, Revised the paper; HZ, Measured IP accumulation, Completed molecular biology experiments; ES, Designed experiments, Carried out Ca2+ flux measurements; TCC, MB, YC, Completed molecular biology experiments; PS, APB, SCB, H-cM, DDC, Performed single-cell Ca2+ microfluorimetry; TXN, Carried out SPA analysis; BC, Measured IP accumulation; MQ, Designed experiments, Carried out SPA analysis; ADC, HMC, Designed experiments, Performed single-cell Ca2+ microfluorimetry, Revised the paper; PM, Designed experiments, Carried out Ca2+ flux measurements, Revised the paper; QRF, Conceived the study, Designed experiments, Collected data, Solved structures, Analyzed structures, Measured IP accumulation, Wrote the paper, Revised the paper

### Author ORCIDs

Qing R Fan, http://orcid.org/0000-0002-9330-0963

## Additional files

### Major datasets

The following datasets were generated:

| Author(s) | Year | Dataset title | Dataset URL | Database, license, and accessibility information |
|---|---|---|---|---|
| Yong Geng, Lidia Mosyak, Igor Kurinov, Hao Zuo, Emmanuel Sturchler, Tat Cheung Cheng, Prakash Subramanyam, Alice P Brown, Sarah C Brennan, Heechang Mun, Martin Bush, Yan Chen, Trang X Nguyen, Baohua Cao, Donald D Chang, Matthias Quick, Arthur D Conigrave, Henry M Colecraft, Patricia McDonald, Qing R Fan | 2016 | Crystal structure of the inactive form of human calcium-sensing receptor extracellular domain | http://www.rcsb.org/pdb/search/structid-Search.do?structureId=5K5T | Publicly available at the RCSB Protein Data Bank (accession no. 5K5T) |
| Yong Geng, Lidia Mosyak, Igor Kurinov, Hao Zuo, Emmanuel Sturchler, Tat Cheung Cheng, Prakash Subramanyam, Alice P Brown, Sarah C Brennan, Heechang Mun, Martin Bush, Yan Chen, Trang X Nguyen, Baohua Cao, Donald D Chang, Matthias Quick, Arthur D Conigrave, Henry M Colecraft, Patricia McDonald, Qing R Fan | 2016 | Crystal structure of the active form of human calcium-sensing receptor extracellular domain | http://www.rcsb.org/pdb/search/structid-Search.do?structureId=5K5S | Publicly available at the RCSB Protein Data Bank (accession no. 5K5S) |

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
