## [Decision Letter]

Thank you for submitting your work entitled "Structural mechanism of ligand activation in human calcium-sensing receptor" for consideration by *eLife*. Your article has been reviewed by three peer reviewers, and the evaluation has been overseen by Ehud Isacoff as a guest Reviewing Editor and Randy Schekman as the Senior Editor. One of the three reviewers has agreed to reveal their identity: Yoshihiro Kubo.

The reviewers have discussed the reviews with one another and the Reviewing Editor has drafted this decision to help you prepare a revised submission.

Summary:

Geng et al. describe the first crystal structures of the calcium-sensing receptor (CaSR), a G protein-coupled receptor (GPCR). These structures of the extracellular domain demonstrate that L-Trp is an orthosteric agonist of the receptor and reveal binding sites for modulating Ca^2+^ and PO_4_^3-^ ions. Structures in four pharmacologically relevant conditions reveal two distinct conformations consistent with a model of an activation rearrangement that is triggered by venus flytrap closure on agonist, which positions the cysteine-rich domains of the subunits in close proximity – consistent with earlier structural and biochemical analysis on other class C GPCRs. The study suggests that positive and negative allosteric modulation by Ca^2+^ and PO_4_^3-^ ions act occurs via modulation of the binding of the orthosteric agonist.

Essential revisions:

1) Evaluate the L-Trp EC_50_ in cell-based assays using different ionic conditions (such as no Ca^2+^ to 10 mM Ca^2+^).

2) To demonstrate the inhibitory effect of PO_4_^3-^ on the response of CaSR to Ca^2+^, compare responses (IP accumulation) to extracellular Ca^2+^ in the presence and absence of PO_4_^3-^, in wt and mutants of the anion binding sites.

3) To demonstrate the potentiating effect of L-Trp on the response to Ca^2+^, compare responses (IP accumulation) to extracellular Ca^2+^ in the presence and absence of L-Trp, in wt and mutants of the L-tryptophan binding sites.

Reviewer #1:

The manuscript by Geng et al. describes the crystal structures of the human calcium-sensing G protein-coupled receptor extracellular domain (ECD). In a nutshell, these structures demonstrate that L-Trp is an orthosteric agonist of the receptor and reveal several binding sites for Ca^2+^ and PO_4_^3-^ ions that, based on mutagenesis data and functional assays, may play a key role in balancing receptor activity.

In particular, the structures were obtained in four different pharmacologically relevant conditions (with or without Ca^2+^ and L-Trp) and revealed only two distinct conformations. Based on the structural information available for the same receptor family (class C GPCRs), the authors propose that these two conformations represent inactive and active states of the receptor. They provide a detailed comparison of both structures to highlight the rearrangement of the ECDs that is necessary for receptor activation. Of note, this is the first structure in the class C family in which the cystein rich domain (CRD) adopt such a conformation in an agonist bound condition. This structure nicely fits the crosslinking data obtained for the metabotropic glutamate receptors in living cells and thus validate the ECD activation model of class C GPCR possessing a CRD domain: an interdomain movement triggered by VFT closure which positions the CRDs in close proximity.

I thus believe this study is of immediate interest to many people in the GPCR field and beyond. Overall the methodology and data are robust and I did not find any flaws that should prohibit its publication.

To conclude, I think it is tempting to speculate that the Ca^2+^ and PO_4_^3-^ ions act respectively as positive and negative allosteric modulators of the CaSR receptor activity by modulating, at least in part, the binding of the orthosteric agonist L-Trp. One key experiment would thus be to evaluate the L-Trp EC_50_ in cell-based assays using different ionic conditions (such as no Ca^2+^ to 10 mM Ca^2+^).

Although this might strengthen the manuscript, I do not believe these data would be critical for publication.

Reviewer #2:

This is an important manuscript reporting the first crystal structure of CaSR. The structure showed interesting features in terms of ligand binding sites. The binding site for Ca^2+^, a main agonist, was different from the canonical ligand binding sites identified in other class C GPCRs, such as mGluR and GABABR. It is noteworthy that this canonical site in CaSR was occupied by L-tryptophan, a putative co-agonist. These features concerning about binding sites and the proposed activation mechanism of CaSR is unique, in comparison to other class C GPCRs. Therefore, this paper will contribute to facilitate the understanding of physiological and biophysical properties of CaSR and class C GPCR. The authors also identified the putative binding sites for PO_4_^3-^, and suggested that PO_4_^3-^ is an antagonist of CaSR for the first time.

I highly evaluate the scientific merits and the general impact of the paper. I judge it is worthy for publication, but only after satisfactory revisions. I have four major comments which require attention.

1) Experiments to demonstrate the inhibitory effect of PO_4_^3-^

The conclusion in the Discussion that " PO_4_^3-^ ions promote the inactive configuration" is rather speculative from the two reasons in the following.

A) There are no direct evidence which shows PO_4_^3-^ promotes the inactivated conformation. The authors showed mutants of the PO_4_^3-^ binding sites have lower IP accumulation in response to extracellular Ca^2+^. I think the data are not sufficient. To demonstrate the inhibitory effect of PO_4_^3-^ on the response of CaSR to extracellular Ca^2+^ conclusively, I would like to request to present data comparing the responses (IP accumulation) to extracellular Ca^2+^ in the presence and absence of PO_4_^3-^, in wt and mutants of the anion binding sites. The expected result is lower response to Ca^2+^ in the presence of PO_4_^3-^ than in the absence, not in mutants but in wt.

B) The authors assume that the electron density represents the presence of PO_4_^3-^ or SO_4_^3-^, because there are no anions in the crystallization solution other than PO_4_^3-^ or SO_4_^3-^. Is there a possibility that carryover endogenous anion(s) other than PO_4_^3-^ is bound to the anion binding site? (similarly to the case of the putative carryover of endogenous amino acid(s) at the L-tryptophan binding site when crystalized in the absence of L-tryptophan).

2) Experiments to demonstrate the potentiating effect of L-tryptophan

I have similar comments to the case of anion effect. To demonstrate the potentiating effect of L-tryptophan on the response of CaSR to extracellular Ca^2+^ conclusively, please present data comparing the responses (IP accumulation) to extracellular Ca^2+^ in the presence and absence of L-tryptophan, in wt and mutants of the L-tryptophan binding sites. The expected result is higher response to Ca^2+^ in the presence of L-tryptophan than in the absence, not in mutants but in wt.

3) Comparison with the mGluR and GABABR

The authors discussed differences and similarities of the CaSR structure to those of mGluR and GABABR. However, it is hard to follow the explanations in the text, without structure images for comparison.

A) The authors mentioned the degree of rotational change of two LB1 interface in CaSR with GABAB and mGluR upon activation (subsection “Common protomer-protomer interactions”, third paragraph). In Figure 3—figure supplement 1, the authors showed two images of CaSR in the resting and activated states. I would like to request to add similar images in the two states of mGluR and GABABR in Figure 3—figure supplement 1, to clarify the discussion of the rotated angle of LB1 interface.

B) The authors described an overall similarity of VFT cleft structure of CaSR and mGluR. They also wrote the tryptophan binding site of CaSR corresponds to the glutamate binding site of mGluR (subsection “Structure of CaSR ECD homodimer”, second paragraph). To clarify these points, please present a superimposed image of the VFT cleft of CaSR and mGluR, in e.g. Figure 1—figure supplement 1.

4) Gd3+ binding site

In the crystal structure of mGluR1, a binding site for Gd3+ was identified at the dimer interface (Tsuchiya et al., PNAS 2002). I would like to have explanations whether or not a similar structure was identified in CaSR.

Reviewer #3:

Geng et al. reported the crystal structures of entire extracellular domain of calcium sensing receptor of calcium free form and loaded forms in the presence of calcium and PO43- ions. This work is of importance for providing insights for the activation mechanism via dimer interface between subunits. There are several major concerns that need be addressed:

1) The crystal structures were determined for CaSR with mutations of either two (N386Q and S402N) or three (N468Q) N-glycosylation sites. However, no study has been carried out on the effects of these mutations on expression and function of CaSR. It is important to provide evidence with experimental data that mutations at these glycosylation sites do not result in alteration of the function and expression.

2) Since mutations could lead to reduction in cell surface expression due to trafficking, it is encouraged to provide confocal images to ensure that the observed change in intracellular IP activity is not due to reduction of protein expression levels on cell surface.

3) The identification of Trp binding site is very important for the activation mechanism. It will be very helpful to report Trp binding affinity of the ECD of CaSR.

4) Extremely high concentrations of three different anion ions were used for crystallization. The role of anion binding is not clear and some of them appear to be non-specific. Among 4 anion-binding sites identified, sites 1-3 are in inactive form crystalized with 3.0 M Li2SO4, 100 mM Tris pH8.5 with mutations N386Q and S402N. Under this condition, SO42- ions are assigned. On the other hand, 1.6 M NaH2PO4, 0.4 M K2HPO4, 100 mM Na2HPO4/citric acid pH4.2, 10 mM Trp 10 mM Ca^2+^ are used for identification of anion sites in the active form of CaSR with three glycosylation mutations. Thus, 2 anion binding sites were modeled to be PO43. What is the binding affinity for these anion ions?

5) The four calcium binding sites shown in Figure 5 either have incomplete coordination or lack of negatively charged residues. These properties are very different from classical calcium binding sites with at least three ligand oxygen atoms from the protein and with electrostatic interactions. The reported site 3 calcium is directly chelated only by one oxygen from hydroxyl side chin of S302. Site 4 is formed by side chain of D234 and mainchain carbonyl oxygen from E231 and G557. How the function results provided in Figure 6 is a mutation G55E that results in a significantly increase of EC50 for IP accumulation and intracellular calcium response. Why have such big alteration of activity for a replacement of another carbonyl oxygen? Since calcium interaction is largely electrostatic, it would make sense to test the mutational effect D234A. The provided mutational studies of I181 m, T100I, and N102I all have significantly reduced maximal amplitudes in both IP and intracellular calcium activities in addition to increased EC50. These results suggest that the surface expression of the mutated protein variants are largely reduced. Thus, the effect for these residues in calcium binding cannot be unambiguously revealed. It is important to provide the results for the mutational effect of these identified ligand residues on direct calcium binding affinity.

6) It is worth to pointing out that at pH4.2 sidechain of Asp and Glu are largely protonated with significantly reduced calcium binding capability. In addition, due to poor solubility of calcium phosphate, the molar phosphate concentrations used in the buffer will result in extremely low concentration of soluble free calcium. Thus, the role of calcium binding in activation revealed in the determined structures is likely to be under estimated due to the condition used for crystallization.

[Editors' note: further revisions were requested prior to acceptance, as described below.]

Thank you for resubmitting your work entitled "Structural mechanism of ligand activation in human calcium-sensing receptor" for further consideration at *eLife*. Your revised article has been favorably evaluated by Randy Schekman (Senior Editor), a Reviewing Editor, and one reviewer.

The manuscript has been improved but there is a remaining issue that needs to be addressed before acceptance, as outlined below:

1) In the point by point response to Reviewer #3, comment #3, the authors stated the following:

"The presence of 2 mM Ca^2+^ did not affect the binding affinity of L-Trp, indicating that Ca^2+^ion is not directly involved in L-Trp recognition. This is consistent with our structural observation that no Ca^2+^ ion is found at the L-Trp-binding site. Nevertheless, Ca^2+^ ion increased the amount of L-Trp bound to CaSR ECD at any given concentration, indicating that extracellular Ca^2+^ enhances the L-Trp-binding site occupancy, possibly by stabilizing the L-Trp-bound and active conformation of CaSR."

The authors should clarify what they mean by this. If calcium stabilized the Trp-bound state it would presumably be seen as an increase in Trp affinity. An increase in amount of radio-Trp that is bound that does not shift the affinity curve would seem to suggest adding to circulation Trp binding sites (i.e. receptors) that were previously not available (from an unavailable/inactive pool?). Or is there another possible explanation? Or does this raise a caveat about the experiment? This should be addressed in the text.

Reviewer #1:

I believe the revised version of the manuscript "Structural mechanism of ligand activation in human calcium-sensing receptor" is addressing the essential revisions that were raised during the first reviewing process. In particular, the authors have addressed the allosteric effect of Ca^2+^ on L-Trp activity (and vice versa).

---

## [Author Response]

*Essential revisions:*

1) Evaluate the L-Trp EC_50_ in cell-based assays using different ionic conditions (such as no Ca^2+^ to 10 mM Ca^2+^).

Following the referees’ suggestion, we have evaluated the L-Trp EC50 at different Ca^2+^ concentrations by monitoring changes in CaSR-mediated intracellular Ca^2+^ response in single cells. In agreement with previous findings, L-Trp can activate CaSR when the extracellular Ca^2+^ concentration is above the threshold level of 0.5 mM. The effect of L-Trp on CaSR was concentration-dependent, with an apparent EC50 of 0.12 ± 0.06 mM when extracellular Ca^2+^ was present at 2.5 mM. The efficacy of L-Trp decreased at lower concentrations of Ca^2+^ that are within the physiological range (L-Trp EC50 = 0.75 ± 0.51 mM at 1.5 mM Ca^2+^), consistent with previous proposal that multiple amino acids need to act in concert to control the function of CaSR. The new data are presented in Figure 4.

2) To demonstrate the inhibitory effect of PO_4_^3-^ on the response of CaSR to Ca^2+^, compare responses (IP accumulation) to extracellular Ca^2+^ in the presence and absence of PO_4_^3-^, in wt and mutants of the anion binding sites.

Following the referees’ suggestion, we have measured the response of CaSR to extracellular Ca^2+^ in the absence and presence of the anion SO_42-_. We chose SO_42-_ to substitute for PO_43-_ because CaSO_4_ is more soluble than CaHPO_4_, the predominant form of calcium phosphate salt at physiological pH. At pH 7.4, approximately 61.3% of the phosphate ions exist in the form of HPO_42-_ and 38.7% are in the form of H_2_PO_4_^-^. The solubility of CaHPO_4_ and Ca(H_2_PO_4_)2 are 1.5 mM and 71 mM, respectively. The solubility of CaSO_4_ is 15 mM.

As predicted by the referee, and consistent with our structural observation, we found that the response of CaSR to Ca^2+^ was slightly lower in the presence of than in the absence SO_4_^2-^. This confirms that anions have a negative allosteric effect on the receptor. The new data are presented in Figure 8.

We have also examined the effects of SO_4_^2-^ on mutants of the anion binding-sites. Our structures revealed that the anion-binding residues R62 at site 1 and R66 at site 3 form hydrogen bonds across the interdomain cleft in the absence of bound anions to stabilize the closed conformation of the receptor. We found that the effect of SO_4_^2-^ on Ca^2+^-induced activation of the R62M mutant was similar to that of the wild-type receptor, while the presence of SO_4_^2-^ had little effect on the activity of the R66H mutant. However, we cannot conclude which anion-binding sites are involved in mediating the inhibitory effect of anions because these mutations are expected to have both positive and negative effects on the receptor function.

A) The mutations R62M and R66H are expected to eliminate the inhibitory effect of anions by preventing their binding to the receptor.

B) These same mutations would also disrupt the formation of hydrogen bonds across the interdomain cleft by R62 and R66, thereby eliminating the interactions that favor receptor activation.

C) Residue R66 is involved in the binding of anion at both sites 2 and 3. Since anion-binding site 2 appears to be crucial for structural integrity of the receptor, disruption of this site by the R66H mutation would negatively affect receptor function.

Author response image 1.Effect of SO_4_^2-^ on Ca^2+^-stimulated IP accumulation in cells transiently expressing wild-type (WT) or mutant CaSR.The mutations R62M and R66H affect arginine residues at anion-binding sites 1 and 3 that form hydrogen bonds formed across the interdomain cleft in the absence of any bound anion.**DOI:**
http://dx.doi.org/10.7554/eLife.13662.023

3) To demonstrate the potentiating effect of L-Trp on the response to Ca^2+^, compare responses (IP accumulation) to extracellular Ca^2+^ in the presence and absence of L-Trp, in wt and mutants of the L-tryptophan binding sites.

Following the referee’s suggestion, we have measured the potentiating effect of L-Trp on the response of wild-type CaSR to extracellular Ca^2+^. During the course of our study, we found that it was very difficult to measure the functional effect of L-Trp through IP accumulation. Instead, we monitored the effect of L-Trp on CaSR-mediated intracellular Ca^2+^ mobilization. As predicted by the referee, Ca^2+^-induced receptor activation is higher in the presence of L-Trp than in the absence. The new data are presented in Figure 4.

We have shown using IP accumulation measurements that most alanine mutants of the L-Trp- binding site abolished Ca^2+^-dependent receptor response in the absence of any amino acid. Therefore, these mutants would not be suitable for the study of the L-Trp effect. For the two mutants that displayed partial receptor activity (S147A and Y218S), we found that they did not respond to 10 mM L-Trp in the presence of 2 mM extracellular Ca^2+^.

Author response image 2.Effect of various concentrations of L-Trp (1, 2, 0, 10 mM) on intracellular Ca^2+^ mobilization in the presence of 2 mM extracellular Ca^2+^.(**A**) Wild-type (WT) CaSR. (**B**) S147A mutant. (**C**) Y218A mutant.**DOI:**
http://dx.doi.org/10.7554/eLife.13662.024

Reviewer #1:

I think it is tempting to speculate that the Ca^2+^ and PO_4_^3-^ ions act respectively as positive and negative allosteric modulators of the CaSR receptor activity by modulating, at least in part, the binding of the orthosteric agonist L-Trp. One key experiment would thus be to evaluate the L-Trp EC_50_ in cell-based assays using different ionic conditions (such as no Ca^2+^ to 10 mM Ca^2+^).

*Although this might strengthen the manuscript, I do not believe these data would be critical for publication.*

Following the referee’s suggestion, we have evaluated the L-Trp EC50 at different Ca^2+^ concentrations by monitoring changes in CaSR-mediated intracellular Ca^2+^ response in single cells. In agreement with previous findings, L-Trp can activate CaSR when the extracellular Ca^2+^ concentration is above the threshold level of 0.5 mM. The effect of L-Trp on CaSR was concentration-dependent, with an apparent EC50 of 0.12 ± 0.06 mM when extracellular Ca^2+^ was present at 2.5 mM. The efficacy of L-Trp decreased at lower concentrations of Ca^2+^ that are within the physiological range (L-Trp EC50 = 0.75 ± 0.51 mM at 1.5 mM Ca^2+^), consistent with previous proposal that multiple amino acids need to act in concert to control the function of CaSR. The new data are presented in Figure 4.

Reviewer #2:

*I highly evaluate the scientific merits and the general impact of the paper. I judge it is worthy for publication, but only after satisfactory revisions. I have four major comments which require attention.*

1) Experiments to demonstrate the inhibitory effect of PO_4_^3-^

*The conclusion in the Discussion that " PO_4_^3-^ ions promote the inactive configuration" is rather speculative from the two reasons in the following.*

*A) There are no direct evidence which shows PO_4_^3-^ promotes the inactivated conformation. The authors showed mutants of the PO_4_^3-^ binding sites have lower IP accumulation in response to extracellular Ca^2+^. I think the data are not sufficient. To demonstrate the inhibitory effect of PO_4_^3-^ on the response of CaSR to extracellular Ca^2+^ conclusively, I would like to request to present data comparing the responses (IP accumulation) to extracellular Ca^2+^ in the presence and absence of PO_4_^3-^, in wt and mutants of the anion binding sites. The expected result is lower response to Ca^2+^ in the presence of PO_4_^3-^ than in the absence, not in mutants but in wt.*

Following the referee’s suggestion, we have measured the response of CaSR to extracellular Ca^2+^ in the absence and presence of the anion SO_4_^2-^. We chose SO_4_^2-^ to substitute for PO_4_^3-^ because CaSO_4_ is more soluble than CaHPO_4_, the predominant form of calcium phosphate salt at physiological pH. At pH 7.4, approximately 61.3% of the phosphate ions exist in the form of HPO_4_^2-^ and 38.7% are in the form of H_2_PO_4_^-^. The solubility of CaHPO_4_ and Ca(H_2_PO_4)2_ are 1.5 mM and 71 mM, respectively. The solubility of CaSO_4_ is 15 mM.

As predicted by the referee, and consistent with our structural observation, we found that the response of CaSR to Ca^2+^ was slightly lower in the presence of than in the absence SO_4_^2-^. This confirms that anions have a negative allosteric effect on the receptor. The new data are presented in Figure 8.

We have also examined the effects of SO_4_^2-^ on mutants of the anion binding-sites. Our structures revealed that the anion-binding residues R62 at site 1 and R66 at site 3 form hydrogen bonds across the interdomain cleft in the absence of bound anions to stabilize the closed conformation of the receptor. We found that the effect of SO_4_^2-^ on Ca^2+^-induced activation of the R62M mutant was similar to that of the wild-type receptor, while the presence of SO_4_^2-^ had little effect on the activity of the R66H mutant. However, we cannot conclude which anion-binding sites are involved in mediating the inhibitory effect of anions because these mutations are expected to have both positive and negative effects on the receptor function.

A) The mutations R62M and R66H are expected to eliminate the inhibitory effect of anions by preventing their binding to the receptor.

B) These same mutations would also disrupt the formation of hydrogen bonds across the interdomain cleft by R62 and R66, thereby eliminating the interactions that favor receptor activation.

Residue R66 is involved in the binding of anion at both sites 2 and 3. Since anion-binding site 2 appears to be crucial for structural integrity of the receptor, disruption of this site by the R66H mutation would negatively affect receptor function.

B) The authors assume that the electron density represents the presence of PO_4_^3-^ or SO_4_^3-^, because there are no anions in the crystallization solution other than PO_4_^3-^ or SO_4_^3-^. Is there a possibility that carryover endogenous anion(s) other than PO_4_^3-^ is bound to the anion binding site? (similarly to the case of the putative carryover of endogenous amino acid(s) at the L-tryptophan binding site when crystalized in the absence of L-tryptophan).

We agree with the referee that it is possible carryover endogenous anion(s) other than PO_43-_ or SO_42-_ is bound at the anion-binding sites. We have added a sentence in the text to mention the caveat.

*2) Experiments to demonstrate the potentiating effect of L-tryptophan*

I have similar comments to the case of anion effect. To demonstrate the potentiating effect of L-tryptophan on the response of CaSR to extracellular Ca^2+^ conclusively, please present data comparing the responses (IP accumulation) to extracellular Ca^2+^ in the presence and absence of L-tryptophan, in wt and mutants of the L-tryptophan binding sites. The expected result is higher response to Ca^2+^ in the presence of L-tryptophan than in the absence, not in mutants but in wt.

Following the referee’s suggestion, we have measured the potentiating effect of L-Trp on the response of wild-type CaSR to extracellular Ca^2+^. As predicted by the referee, Ca^2+^-induced receptor activation is higher in the presence of L-Trp than in the absence. The new data are presented in Figure 4.

We have shown using IP accumulation measurements that most alanine mutants of the L-Trp- binding site abolished Ca^2+^-dependent receptor response in the absence of any amino acid. Therefore, these mutants would not be suitable for the study of the L-Trp effect. For the two mutants that displayed partial receptor activity (S147A and Y218S), we found that they did not respond to 10 mM L-Trp in the presence of 2 mM extracellular Ca^2+^.

*3) Comparison with the mGluR and GABABR*

*The authors discussed differences and similarities of the CaSR structure to those of mGluR and GABABR. However, it is hard to follow the explanations in the text, without structure images for comparison.*

*A) The authors mentioned the degree of rotational change of two LB1 interface in CaSR with GABAB and mGluR upon activation (subsection “Common protomer-protomer interactions”, third paragraph). In Figure 3—figure supplement 1, the authors showed two images of CaSR in the resting and activated states. I would like to request to add similar images in the two states of mGluR and GABABR in Figure 3—figure supplement 1, to clarify the discussion of the rotated angle of LB1 interface.*

Following the referee’s recommendation, we have added images of the two states of mGluR1 and GABAB receptor comparing the rotated angle of the LB1-LB1 interface in each receptor system (Figure 3—figure supplement 1).

B) The authors described an overall similarity of VFT cleft structure of CaSR and mGluR. They also wrote the tryptophan binding site of CaSR corresponds to the glutamate binding site of mGluR (subsection “Structure of CaSR ECD homodimer”, second paragraph). To clarify these points, please present a superimposed image of the VFT cleft of CaSR and mGluR, in e.g. Figure 1—figure supplement 1.

We agree with the referee and have added a superimposed image of the agonist-binding cleft of CaSR and mGluR1 in Figure 4—figure supplement 1.

*4) Gd3+ binding site*

In the crystal structure of mGluR1, a binding site for Gd3+ was identified at the dimer interface (Tsuchiya et al., PNAS 2002). I would like to have explanations whether or not a similar structure was identified in CaSR.

We think the referee’s question is very interesting, and have superimposed the structures of CaSR and mGluR to compare this region of the dimer interface. As shown in Figure 12, the Gd^3+^ ion in the mGluR1 structure is coordinated by two acid residues, E238 and D242 from each subunit. The corresponding residues in the CaSR structure are R220 and E224. The basic residue R220 would not be compatible with the binding of a cation to CaSR at this location.

Author response image 3.Superposition of the CaSR and mGluR1 structures in the region of the Gd^3+^-binding site in mGluR1 structure.**DOI:**
http://dx.doi.org/10.7554/eLife.13662.025

Reviewer #3:

*Geng et al. reported the crystal structures of entire extracellular domain of calcium sensing receptor of calcium free form and loaded forms in the presence of calcium and PO43- ions. This work is of importance for providing insights for the activation mechanism via dimer interface between subunits. There are several major concerns that need be addressed:*

*1) The crystal structures were determined for CaSR with mutations of either two (N386Q and S402N) or three (N468Q) N-glycosylation sites. However, no study has been carried out on the effects of these mutations on expression and function of CaSR. It is important to provide evidence with experimental data that mutations at these glycosylation sites do not result in alteration of the function and expression.*

We agree with the referee that it is important to provide evidence that the glycosylation mutations introduced into our expression constructs do not alter the function of the receptor. It has been reported previously that glycosylation is not required for signal transduction of CaSR (Ray et al., JBC, 1998, 273, 34558-34567). Following the referee’s suggestion, we have also generated two full-length CaSR constructs for functional analysis in mammalian cells, one carrying the two glycosylation-site mutations N386Q and S402N, and the other carrying the additional mutation N468Q. We found that Ca^2+^-induced IP accumulation was statistically the same for the wild-type and mutant receptors, indicating that these glycosylation mutations did not alter CaSR function (Figure 1—figure supplement 1).

On the other hand, previous studies of full-length CaSR in mammalian cells indicate that disruption of the glycosylation sites impairs proper processing and expression of the receptor at the cell surface (Ray et al., JBC, 1998, 273, 34558-34567). In agreement with these findings, we found that expression of CaSR ECD in insect cells was essentially eliminated when more than three glycosylation sites were mutated. After testing mutations at each of the potential N-linked glycosylation sites individually and in various combinations, we chose the constructs reported in this study to maximize the reduction of carbohydrate content while maintaining a sufficient expression level for structural analyses.

*2) Since mutations could lead to reduction in cell surface expression due to trafficking, it is encouraged to provide confocal images to ensure that the observed change in intracellular IP activity is not due to reduction of protein expression levels on cell surface.*

We appreciate the referee’s suggestion. Unfortunately, we do not have any experience with confocal imaging. We did measure the cell surface expression levels of wild-type and mutant CaSR by on-cell western analysis, which was described in the Methods section. Most of the mutants in our study were expressed on the cell surface at levels comparable to that of the wild- type receptor. The exceptions were R66H, R69E, I81M, T100I, N102I, and S417L, which reduced the surface expression of the mutant receptors to approximately 70-75% of the wild-type level. Nevertheless, the reduction in the protein expression level on cell surface could not account for the much greater decrease in intracellular IP accumulation for each of these mutants. We therefore concluded that the mutations affected receptor function.

*3) The identification of Trp binding site is very important for the activation mechanism. It will be very helpful to report Trp binding affinity of the ECD of CaSR.*

We agree with the referee, and have determined the binding affinity of L-Trp for CaSR ECD by scintillation proximity assay using radiolabeled [^3^H]-Trp. In the absence of Ca^2+^, the binding of [^3^H]-L-Trp to CaSR ECD was inhibited by nonradioactive L-Trp with a half-maximal inhibitory concentration (IC_50_) of 2.1 ± 0.7 mM. The presence of 2 mM Ca^2+^ did not affect the binding affinity of L-Trp, indicating that Ca^2+^ ion is not directly involved in L-Trp recognition. This is consistent with our structural observation that no Ca^2+^ ion is found at the L-Trp-binding site. Nevertheless, Ca^2+^ ion increased the amount of L-Trp bound to CaSR ECD at any given concentration, indicating that extracellular Ca^2+^ enhances the L-Trp-binding site occupancy, possibly by stabilizing the L-Trp-bound and active conformation of CaSR.

*4) Extremely high concentrations of three different anion ions were used for crystallization. The role of anion binding is not clear and some of them appear to be non-specific. Among 4 anion-binding sites identified, sites 1-3 are in inactive form crystalized with 3.0 M Li2SO4, 100 mM Tris pH8.5 with mutations N386Q and S402N. Under this condition, SO42- ions are assigned. On the other hand, 1.6 M NaH2PO4, 0.4 M K2HPO4, 100 mM Na2HPO4/citric acid pH4.2, 10 mM Trp 10 mM Ca^2+^ are used for identification of anion sites in the active form of CaSR with three glycosylation mutations. Thus, 2 anion binding sites were modeled to be PO43-. What is the binding affinity for these anion ions?*

We agree with the referee that it would be informative to determine the binding affinities of the anions to CaSR. We tried to quantify the binding of PO_43-_ ions to purified CaSR ECD protein using microscale thermophoresis, but were unsuccessful. Several factors may have contributed to the difficulties we encountered. (1) The anions at different sites may have overlapping binding affinity ranges that cannot be easily differentiated. (2) Some sites such as site 2 may already be occupied by endogenous anions. (3) Low-affinity anion-receptor interactions at some sites are difficult to detect.

*5) The four calcium binding sites shown in Figure 5 either have incomplete coordination or lack of negatively charged residues. These properties are very different from classical calcium binding sites with at least three ligand oxygen atoms from the protein and with electrostatic interactions. The reported site 3 calcium is directly chelated only by one oxygen from hydroxyl side chin of S302. Site 4 is formed by side chain of D234 and mainchain carbonyl oxygen from E231 and G557. How the function results provided in Figure 6 is a mutation G55E that results in a significantly increase of EC50 for IP accumulation and intracellular calcium response. Why have such big alteration of activity for a replacement of another carbonyl oxygen? Since calcium interaction is largely electrostatic, it would make sense to test the mutational effect D234A.*

We appreciate the referee’s observation that the coordination of Ca^2+^ ions at all four binding sites in CaSR appear different from classical Ca^2+^-binding sites. It is possible that water molecules play a role in coordinating the bound Ca^2+^ ions, but could not all be seen in the structures due to disordering or the modest resolution of the structures.

The alteration in receptor activity of the G557E mutant may have resulted from a change in the conformation of the loop containing residue 557 such that the carbonyl oxygen is no longer optimally positioned to coordinate the binding of a Ca^2+^ ion at this site. Nevertheless, the effect of the G557E mutation is not as substantial as some of the other Ca^2+^-binding site mutations including I81M, T100I and N102I, which nearly abolished Ca^2+^-induced receptor activation. The EC_50_ for Ca^2+^-induced IP accumulation increased approximately two fold from 1.6 mM for the wild-type receptor to 3.4 mM for the G557E mutant, while the Emax of the receptor response remained essentially the same.

Following the referee’s suggestion, we have tested the mutation D234A, and found that its effect on receptor activity was negligible, as shown by the IP accumulation dose response curves for wild-type receptor and the D234A mutant (Figure 13). This may be due to the fact that the side chain of D234 is somewhat flexible, and does not contribute to the binding of Ca^2+^ ion at site 4 as much as the backbone carbonyl oxygen atoms of E231 and G557.

Author response image 4.Dose-dependent Ca^2+^-stimulated IP accumulation in cells transiently expressing the D234A mutant CaSR.**DOI:**
http://dx.doi.org/10.7554/eLife.13662.026

*The provided mutational studies of I181 m, T100I, and N102I all have significantly reduced maximal amplitudes in both IP and intracellular calcium activities in addition to increased EC50. These results suggest that the surface expression of the mutated protein variants are largely reduced. Thus, the effect for these residues in calcium binding cannot be unambiguously revealed. It is important to provide the results for the mutational effect of these identified ligand residues on direct calcium binding affinity.*

The cell surface expression levels of the mutants I81M, T100I, N102I were approximately 70- 75% of the wild-type level based on on-cell western measurements. However, each of these mutations nearly abolished Ca^2+^-dependent receptor response, indicating that their effects on calcium binding and receptor function could not be explained by the reduction in cell surface expression alone.

We agree with the referee that it would be very informative to measure the direct binding affinity of Ca^2+^ at each site, and the mutational effects of Ca^2+^-binding residues. We tried two different approaches to determine the Ca^2+^-binding affinity to wild-type CaSR ECD, including microscale thermophoresis and scintillation proximity assay. Unfortunately, we were not successful possibly for the same reasons that we had difficulties measuring the affinities of anions. (1) The Ca^2+^ ions at different sites may have overlapping binding affinity ranges that cannot be easily differentiated. (2) Some sites such as site 2 may already be occupied by endogenous Ca^2+^. (3) Low-affinity Ca^2+^-receptor interactions at some sites are difficult to detect.

*6) It is worth to pointing out that at pH4.2 sidechain of Asp and Glu are largely protonated with significantly reduced calcium binding capability. In addition, due to poor solubility of calcium phosphate, the molar phosphate concentrations used in the buffer will result in extremely low concentration of soluble free calcium. Thus, the role of calcium binding in activation revealed in the determined structures is likely to be under estimated due to the condition used for crystallization.*

We think the referee raised an important question. However, we do not think that the role of calcium binding in activation as revealed by the determined structures is likely to be underestimated for two reasons.

1) Based on the following calculation, the proton concentration at pH4.2 is approximately 0.1 mM, much lower than the concentration of free Ca^2+^ (10 mM). Therefore, we think it is more likely for the sidechains of Asp and Glu residues to be bound to Ca^2+^ ion than protonated.

pH = -log[H^+^] = 4.2

[H^+^] = 1 x 10^-4.2^ M ≈ 0.1 mM

2) We think the concentration of soluble free calcium under the pH 4.2 crystallization condition should remain at 10 mM because most of the phosphate ions are in the form of H_2_PO_4_^-^, and the solubility of Ca(H_2_PO_4)2_ is approximately 71 mM.

Three equilibrium reactions exist between H_3_PO_4_, H_2_PO_4_^-^, HPO_4_^2-^, and PO_4_^3-^

EquilibriumConstantValueEquilibriumK_a1_7.5 x 10^-3^H_3_PO_4_ < H^+^ + H_2_PO_4_^-^K_a2_6.2 x 10^-8^H_2_PO_4_^-^ < H^+^ + HPO_4_^2-^K_a3_4.8 x 10^-13^HPO_4_^2-^ < H^+^ + PO_4_^3-^

H_3_PO_4_ < H^+^ + H_2_PO_4_^-^

pK_a1_ = pH - log([H_2_PO_4_^-^]/[H_3_PO_4_])

-log(7.5 x 10^-3^) = 4.2 - log([H_2_PO_4_^-^]/[H_3_PO_4_])

[H_2_PO_4_^-^]/[H_3_PO_4_] = 10^2.1^

H_2_PO_4_^-^ < H^+^ + HPO_4_^2-^

pK_a2_ = pH - log([HPO_4_^2-^]/[H_2_PO_4_^-^])

-log(6.2 x 10^-8^) = 4.2 - log([HPO_4_^2-^]/[H_2_PO_4_^-^])

[H_2_PO_4_^-^]/[HPO_4_^2-^] = 10^3^

HPO_4_^2-^ < H^+^ + PO_4_^3-^

pK_a3_ = pH - log([PO_4_^3-^]/[HPO_4_^2-^])

-log(4.8 x 10^-13^) = 4.2 - log([PO_4_^3-^]/[HPO_4_^2-^])

[HPO_4_^2-^]/[PO_4_^3-^] = 10^8.1^

[H_2_PO_4_^-^]/[PO_4_^3-^] = 10^11.1^

Therefore, more than 99% of the phosphate ions exist in the form of H_2_PO_4_^-^ at pH4.2.

[Editors' note: further revisions were requested prior to acceptance, as described below.]

*Thank you for resubmitting your work entitled "Structural mechanism of ligand activation in human calcium-sensing receptor" for further consideration at eLife. Your revised article has been favorably evaluated by Randy Schekman (Senior Editor), a Reviewing Editor, and one reviewer.*

*The manuscript has been improved but there is a remaining issue that needs to be addressed before acceptance, as outlined below:*

*1) In the point by point response to Reviewer #3, comment #3, the authors stated the following:*

*"The presence of 2 mM Ca^2+^ did not affect the binding affinity of L-Trp, indicating that Ca^2+^ ion is not directly involved in L-Trp recognition. This is consistent with our structural observation that no Ca^2+^ ion is found at the L-Trp-binding site. Nevertheless, Ca^2+^ ion increased the amount of L-Trp bound to CaSR ECD at any given concentration, indicating that extracellular Ca^2+^ enhances the L-Trp-binding site occupancy, possibly by stabilizing the L-Trp-bound and active conformation of CaSR."*

*The authors should clarify what they mean by this. If calcium stabilized the Trp-bound state it would presumably be seen as an increase in Trp affinity. An increase in amount of radio-Trp that is bound that does not shift the affinity curve would seem to suggest adding to circulation Trp binding sites (i.e. receptors) that were previously not available (from an unavailable/inactive pool?). Or is there another possible explanation? Or does this raise a caveat about the experiment? This should be addressed in the text.*

We think referee #3 has raised a valid criticism. As pointed out by referee #1 below, we cannot conclude based on our [^3^H]-L-Trp binding competition studies alone that the binding affinity of L-Trp is the same with or without Ca^2+^. Given our structural observation that Ca^2+^ stabilizes the active conformation of the receptor, we agree that Ca^2+^ would most likely enhance the binding affinity of L-Trp since the interaction of L-Trp with an active receptor is expected to be more extensive than with an inactive receptor. In an active receptor, L-Trp would contact both the LB1 and LB2 domains. L-Trp may still bind to the open cleft in an inactive receptor, but its interaction with the receptor will be limited to the LB1 domain, which would result in lower affinity. The binding of glutamate to an open cleft has previously been observed for mGluR1 (Kunishima et al., Nature 2000, 407, 971-977).

Unfortunately, as indicated by referee #1, it would be difficult for us to measure the binding affinity of L-Trp in a saturation binding experiment since the concentration of [^3^H]-L-Trp required would be ≥10 mM, and a relatively low specific radioactivity of the commercially available radio-labeled ligand may preclude such experiments. Our competition experiments in the current study were performed at 0.1 mM [^3^H]-L-Trp, a concentration that is about 20-fold below the determined half-maximal inhibitory concentration (IC_50_) of non-labeled L-Trp for replacing radiolabeled L-Trp.

In addition, the presence of Ca^2+^ may affect the on- and off-rate of L-Trp binding to CaSR ECD. Specifically, L-Trp may have a slower off-rate when bound to the active conformation. Therefore, the higher amount of L-Trp binding observed with Ca^2+^ may reflect the different L-Trp binding affinity and kinetics exhibited by the inactive and active receptors. We have modified our text to include these explanations and to indicate that additional studies would be needed to characterize the role of L-amino acid binding to the orthosteric agonist site of CaSR.